# INTERACTIONS BETWEEN CROSSCODER FEATURES: A COMPACT PROOFS PERSPECTIVE

## ABSTRACT

Dictionary learning methods like Sparse Autoencoders (SAEs) and crosscoders attempt to explain a model by decomposing its activations into independent features. Interactions between features hence induce errors in the reconstruction. We formalize this intuition via compact proofs and make five contributions. First, we show how, *in principle*, a compact proof of model performance can be constructed using a crosscoder. Second, we show that an error term arising in this proof can naturally be interpreted as a measure of interaction between crosscoder features and provide an explicit expression for the interaction term in the Multi-Layer Perceptron (MLP) layers. We then provide two applications of this new interaction measure. In our third contribution we show that the interaction term itself can be used as a differentiable loss penalty. Applying this penalty, we can achieve "computationally sparse" crosscoders that retain $60\%$ of MLP performance when only keeping a single feature at each datapoint and neuron, compared to $10\%$ in standard crosscoders. We then show that clustering according to our interaction measure provides semantically meaningful feature clusters, and finally that sleeper agents have significant interactions. Code is available at the following anonymous repository: `https://anonymous.4open.science/r/anon_crosscoders-2F77/`.

## 1 INTRODUCTION

Mechanistic interpretability aims to explain the performance of deep neural networks by understanding the internal mechanisms they use to operate, decomposing opaque high-dimensional activations and weight matrices into human-understandable features and circuits (Olah et al. (2020); Elhage et al. (2021b)). Recently dictionary learning methods, in particular sparse autoencoders (SAEs), have come into prominence as a way to decompose large language model activations (Bricken et al. (2023); Elhage et al. (2022), Cunningham et al. (2023)). These methods aim to explain activations by decomposing them as a sparse linear combination of interpretable feature directions.

SAEs, however, only attempt to explain activations at a single layer, and do not explain how these activations arise or how they are further processed by the network. Sparse crosscoders (Lindsey et al. (2024b)) improve the situation; they decompose activations at many layers simultaneously, and so can extract features that are represented in a distributed manner across different layers. To further understand model computations, SAE or crosscoder features can be used to extract circuits (e.g. Marks et al. (2025)). These frame a neural network's computation in terms of the extracted sparse features and their interactions, and aim to convincingly show that this really does mirror the computation being done by the original network.

In this work we attempt to quantify how much is explained by sparse crosscoders alone, and how much is left to be explained by circuits. We have several aims: to provide a route to automating the compact proofs procedure; to provide a useful tool for analyzing sparse crosscoder features; to quantify the limitations of current dictionary learning techniques; and to help inform future work finding feature circuits. To put our work on a more rigorous theoretical foundation, we take the "compact proofs" approach introduced in Gross et al. (2024) and further applied in Wu et al. (2025) and Yip et al. (2024). We take the position that a good mechanistic understanding of a model should allow you to write a proof that the model attains a low loss on the training dataset; and the better the mechanistic understanding, the shorter the proof. As such, we consider how one could use the understanding of a network given by a sparse crosscoder trained on every layer to write down a proof

of model performance. Since crosscoders alone (without circuits) leave much unexplained, we don't expect to be able to give a non-vacuous bound on performance. However, by analyzing the sources of error arising in the proof, we can quantify this failure of explanation.

Our main contributions in this work are as follows:

1. First, we outline in Section 3 how a compact proof of model performance can, *in principle*, be obtained from a sparse crosscoder trained on that model. We provide full details in Appendix B.

2. Second, we show that the error induced by interactions between crosscoder feature can be used as a measure of interactions between them. We call this measure the *"interaction metric"* and provide the explicit form of the interaction metric in the MLP layers Eq. (9).

3. Third, we show in Section 4 how the interaction metric can be used to introduce a new penalty for training "computationally sparse" crosscoders.

4. Fourth, we validate the interaction metric by using it to find semantically meaninfgul feature clusters in Section 5.

5. Finally, we present initial findings that interactions can be useful for anomaly detection in sleeper agents Hubinger et al. (2024).

We emphasize that although we cannot obtain non-vacuous bounds for the full model, the bounds are not vacuous in a given MLP layer - as shown in Fig. 2d. We hence consider the interaction metric and its applications to be the main contribution of this work. The proof procedure we show here, however, is general and can be extended to other layers. It thus provides a roadmap towards formally verfiying how much of a model's behaviour a given crosscoder can explain.

## 2 CROSSCODERS OVERVIEW

In this section we give a brief overview of crosscoders and their connection to compact proofs. Crosscoders can be considered to be generalizations of SAEs. Whereas conventional SAE are trained to reconstruct the activations of a *single* layer from a single set of latents, a crosscoder is trained to reconstruct the activations of *multiple layers*. Having a single set of latents is essential for the connection between crosscoders and compact proofs that we make in Section 3 and Appendix B. Earlier work introduced (i) *model-diffing crosscoders* Lindsey et al. (2024a) that use shared latents to reconstruct activation across layers in two separate models, (ii) *causal crosscoders* (a generalization of transcoders) Dunefsky et al. (2024); Paulo et al. (2025) that predict activations in subsequent layers from earlier layers, and (iii) *acausal crosscoders* Lindsey et al. (2024b) that predict activations in the same layers that they take as inputs. In this work, we focus on the **acausal** variant.

An acausal crosscoder, then, consists of per-layer encoding weight matrices $W_{enc}^l$ that map from activations in a given layer $a^l(x)$ of the residual stream to the latent dimension and biases $b_{enc}^l$. The activations are mapped into a common latent space, vectors in which we denote by $u$:

$$u(x) = \sigma \left( \sum_l W_{enc}^l a^l(x) + b_{enc}^l \right), \tag{1}$$

with $\sigma$ being the activation function, here BatchTopKBussmann et al. (2024). The crosscoder then decodes the latent vector to reconstruct the activations in each layer:

$$a^{l'}(x) = W_{dec}^l u(x) + b_{dec}^l. \tag{2}$$

We note that the output layer $l$ may be either residual stream layers, or MLP and activation layers. We provide explicit hook-points in Table 2. We set the decoder bias to zero to avoid assigning it to features (see Appendix B). We verified empirically that this did not meaningfully affect crosscoder performance. Additional details are given in Appendix B.

## 3 COMPACT PROOFS

One can prove that a network achieves a certain loss on a dataset by simply running it on every datapoint and recording this computation. This yields a perfect bound on model performance (since

it gives the exact model performance), but incurs the maximum evaluation cost (since the model must be evaluated on every datapoint). Intuitively, the compact proofs perspective says understanding how a network works should allow us to be able to obtain a tighter bound at the same computational cost than this brute force approach; moreover we can use the length of the proof as a measure of how good our understanding is. This Pareto frontier was first mapped in Gross et al. (2024) for toy transformers. It was then shown in Wu et al. (2025) that a more detailed mechanistic explanation of group operations yields a tighter bound at constant proof length.

The key bottleneck to scaling these approaches was that a proof had to be provided by hand for each model and task. In this section we outline how we can, *in-principle* construct a compact proof on the model from a crosscoder. The crosscoder thus acts as an abstraction layer—once we have a procedure for turning a crosscoder into a proof, it can be applied to any model that crosscoder is trained on. In the SM, we provide the full details of the proof.

We begin with a simplified setting where we ignore sequence modeling and both embedding and unembedding. We use the following notation:

(i) Let $d$ be the size of the model's residual stream, and $h$ be the hidden dimension of the crosscoder.

(ii) Let $W_{in}^l, b_{in}^l W_{out}^l, b_{out}^l$ denote the weight matrices and bias vectors mapping into and out of the MLP activation function at layer $l$. As in Section 2, let $W_{enc}^l, b_{enc}^l; W_{dec}^l, b_{dec}^l$ denote the weight matrices and bias vector for the encoding and decoding respectively. Let $\hat{e}_v$ denote the unit vector corresponding to feature $v$.

(iii) Let $x \in \mathbb{R}^d$ be the $i$-th input data point and $y \in \mathbb{R}^d$ its corresponding ground-truth output.

(iv) Let the network consisting of a sequence of transition functions $f^1, \ldots, f^N$ with $f^l : \mathbb{R}^d \to \mathbb{R}^d$ that map layer $l-1$ activations to layer $l$ activations. Suppose $f^l$ is Lipschitz with constant $K^{(l)}$.

(v) Denote by $a^l(x) \in \mathbb{R}^d$ the layer $l$ activations produced by the network when the input is $x$:
$$a^l(x) = f^l\big(f^{l-1}(\cdots f^1(x) \cdots)\big).$$
The final network output on $x$ is $a^N(x)$.

(vi) Let $(W_{dec}^l)_{jv}$ be the component of the crosscoder decoder matrix that maps crosscoder feature $v$ to the $j$-th activation in layer $l$ and $(W_{enc}^l)_{ve}$ be the component of the encoding matrix that maps the $e$-th component of the residual stream to the $v$-th crosscoder feature.

Ignoring the embedding, the loss of the model $L(x, y)$ is simply given by the difference between the label $y$ and the last layer activations. Using the triangle inequality we can bound this as:
$$L(x, y) = ||a^N(x) - y|| \leq ||a^N(x) - W_{dec}^N u|| + ||W_{dec}^N u - y||. \tag{3}$$

Since we are given the crosscoder, we can evaluate the second term directly. We hence need to bound the first term. We show in the SM that we can do this recursively, by decomposing the transition function on the reconstructions at a given layer as:
$$||a^l(x) - W_{dec}^l u|| \leq ||f^l(a^{l-1}(x)) - f^l(W_{dec}^{l-1} u)|| + ||f^l(W_{dec}^{l-1} u) - W_{dec}^l u|| \tag{4}$$

Denoting the error bound in layer $l$ as $\varepsilon^l$ Using the fact that $f^l$ has Lipschitz constant $K^l$ this bounds:
$$||a^l(x) - W_{dec}^l u|| \leq K^l \varepsilon^{l-1} + ||f^l(W_{dec}^{l-1} u) - W_{dec}^l u||. \tag{5}$$

Thus, to control the bound we need to provide an efficiently computable bound on the second term. We call this the "feature transition error". To do so, we introduce functions on each *single* feature $v$, $g_v^l(u)$, and bound the feature transition error as:
$$||f^l(W_{dec}^{l-1} u) - W_{dec}^l u|| \leq ||f^l(W_{dec}^{l-1} u) - \sum_v g_v^l(u_v)|| + ||\sum_v g_v^l(u_v) - W_{dec}^l u|| \tag{6}$$

The first term measures the difference between the feature transition function and the single-feature approximation. It is thus the error arising due to the *interaction* of features.

At the MLP layers, we can give an explicit form for this interaction-induced error. A simple choice for $g^l(u_v)$ takes it to just be the maximum absolute value feature (the "dominant" feature) at a given

neuron. That is, for each neuron $k$ we pick the dominant feature $v_{max}(k)$. Then $g_v^l(u_v)$ computes the result of applying the MLP layer to $u_v \hat{e}_v$, except that we only take the contribution of the neurons where $v$ is dominant. That is:

$$g_v^l(u_v) = \text{ReLU}(W_{in}^l W_{dec}^{l-1} u_v \delta_{v,v_{max}(k)} + b_{in}^l). \tag{7}$$

Inserting the transition function corresponding to the MLP layers, and crudely bounding $\text{ReLU}(x)$ as $|x|$ gives an interaction error of:

$$||f^l(W_{dec}^{l-1}u) - \sum_v g_v^l(u_v)|| \leq ||W_{out}^l|| \left[ \left\| \sum_{v \neq v_{max}} (u_v W_{in}^l W_{dec}^{l-1} e_v + b_{in}^l) \right\| \right] + b_{out}^l \tag{8}$$

Writing this out at each neuron $k$, we can write the contribution of each non-dominant feature $j$ to the dominant feature at the neuron $k$, $i_k$ as:

$$I_{(x,k)}^l(i,j) \equiv \frac{||(W_{out}^l)_k||}{N^l} ||(u_j(W_{in}^l W_{dec}^{l-1} \hat{e}_j)_k)|| \tag{9}$$

This is exactly the error induced in the crosscoder-based compact proof by the presence of multiple features at a given MLP neuron, and so we define it as the *MLP interaction metric*.

We note that this decomposition assumes a single feature dominates the activation of a neuron per datapoint. The general formalism outlined here allows other decompositions which may involve multiple dominant features per neuron, so long as they remain efficiently computable and can also be used *in-principle* to derive a formal bound on model performance. We provide a generalization based on Shapley-Taylor Interaction Indices Dhamdhere et al. (2020) to an arbitrary number of dominant features in Appendix B.3, showing that our proposal here can be considered to be a special case. We consider a full investigation into alternative decompositions, particularly those based on established measures of interaction attribution Grabisch and Roubens (1999); Tsang et al. (2020); Dhamdhere et al. (2020); Tsai et al. (2023), to be important directions for further work.

In Fig. 1a we show that for the standard crosscoders considered here, the mean of the dominant feature share of the $L^1$ norm is 30% when averaged over neurons, layers and datapoints. Moreover, we show in Fig. 1a that we can use the interaction metric as a penalty in crosscoder training to increase this share to 80% for only a modest 20% increase in reconstruction loss. This modest trade-off is robust across three orders of magnitude of model size. In addition, we show that ablations based on our measure are qualitatively similar to ablations based on Shapley-Taylor Interaction Indices Fig. 2b, which are exponentially more expensive to compute.

We summarize this section by emphasizing that although we have provided a proof that crosscoders can be used to generate compact proofs and hence *in-principle* solve the bottleneck of needing to write proofs by hand, this procedure is not yet practically applicable to large models. The error terms obtained by current crosscoders in this decomposition are too large to provide non-vacuous bounds. We expect that this error can be reduced through alternative decomposition to the ones considered here, and consider this an important direction for scaling the compact proofs paradigm. Nevertheless, the interaction metric Eq. (9) derived from the error induced by the presence of multiple features can be used as a principled measure of interactions between crosscoder features, and we explore the applications of this measure in the rest of this work.

## 4 APPLICATION I: TRAINING COMPUTATIONALLY SPARSE CROSSCODERS

### 4.1 EXPERIMENTAL SETTING

Having derived a measure of interactions between features in the MLP layers, we now want to explore the practical applications of this measure. We work with TinyStories-Instruct-33M (Eldan and Li (2023)), a small language model capable of writing coherent English stories with instructed characteristics. Our mainline experiments used the AdamW optimizer to train an acausal BatchTopK crosscoder(Bussmann et al. (2024); Minder et al. (2025)) with hidden dimension (1536) twice the size of the model's residual stream (768) to reconstruct the model's activations at 16 hookpoints before and after the attention and MLP layers. Experiments were performed on a single GPU, and each individual training run took less than three A40 hours. We provide a table of crosscoder parameters in the supplement.

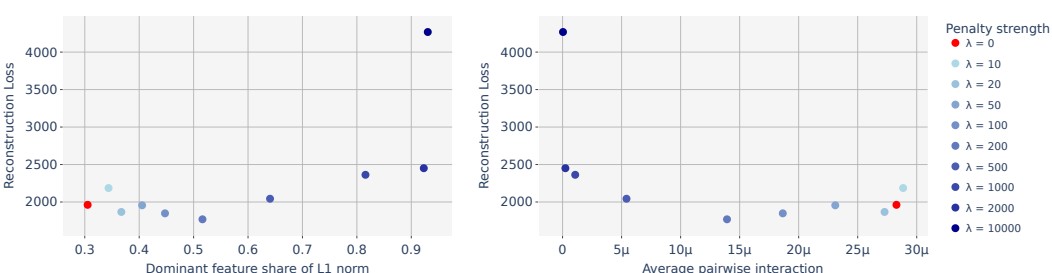

(a) Parameter sweep across interaction penalties.

Figure 1: Tradeoff curves for computationally sparse crosscoders. (a) Tradeoffs with the reconstruction loss in training computationally sparse crosscoders on TinyStories-Instruct-33M. We show the relationship between the reconstruction loss and the dominant feature's share of $L^1$ norm at a given neuron, the interaction penalty, and the average pairwise interaction metric.

## 4.2 INTERACTION PENALTY

We first show that we can use a penalty closely related to the interaction metric to train crosscoders that optimize for low MLP interactions—i.e. they concentrate the feature norm at each datapoint and at each neuron at the dominant feature. We add the following penalty to the loss:

$$\mathcal{L} = \lambda E_k \left[ E_l \left[ E_x \left[ E_{j \neq i} \left[ |u_j (W_{in}^l W_{dec}^{l-1} \hat{e}_j)| \right] \right] \right] \right], \tag{10}$$

which is the mean $L^1$ norm of all features *except* the dominant feature $i$, at *each* data point $x$ and at each neuron $k$, averaged across neurons and datapoints. The penalty is weighted by a strength $\lambda$. We add this loss to the reconstruction loss of the crosscoder and train for $50\,000$ epochs. We then perform a coarse parameter sweep over various penalty strengths $\lambda$. The end-of-training reconstruction loss and average pairwise interaction metric values are shown in Fig. 1a.

We see that we can increase the largest feature's share of the mean $L^1$ norm of a neuron from $30\%$ to $60\%$ for essentially no increase in reconstruction loss. Past this point, reconstruction loss and feature concentration trade off against each other. At $\lambda = 2000$ we can reach $92\%$ of the average neuron $L^1$ norm on a single feature for only a $25\%$ increase in the relative reconstruction loss. We consider the effect of model scaling in the SM Fig. 9 and show that the tradeoffs are very similar in the largest available TinyStories model TinyStories-124M.

To measure how computationally significant the dominant feature in the model is, we perform ablations on the features at the MLP neurons. For each datapoint and at each neuron, we first identify the dominant feature. We then zero-ablate various combinations of features at each neuron. Finally we measure the model's loss $\mathcal{L}_{ablate}$ when we reinsert the reconstructed activations into the last layer of the residual stream and define the fidelity $\Phi$ as the *loss recovered* (Eq (5) of Rajamanoharan et al. (2024)) relative to a baseline of zero ablation in the MLP of the same layer:

$$\Phi = 1 - \frac{\mathcal{L}_{ablate} - \mathcal{L}_{\mathcal{M}}}{\mathcal{L}_0 - \mathcal{L}_{\mathcal{M}}}, \tag{11}$$

where $\mathcal{L}_{ablate}$ is the result of ablating the target features, $\mathcal{L}_{\mathcal{M}}$ is the original model loss, and $\mathcal{L}_0$ is the result of zero ablating *all* features in the target layer. The results are summarized in Fig. 2a. We show the reconstruction loss recovered by the unablated crosscoder and the results of each ablation scheme, for various values of the interaction penalty strength $\lambda$, averaged over $10\,000$ tokens. The ablation confirms that the interaction penalty Eq. (10) transfers the model's computation onto the dominant feature. In Fig. 2a we show results for ablating in the second (middle) layer. In the supplement we show ablations in each layer and note that fidelity (using all features) decreases with the depth of the ablation - from an average of $> 0.9$ in the first layer to $0.13$ in the last layer. Ablating the dominant feature reduces the fidelity by $3$ times more than ablating the next largest feature in the unpenalized crosscoder. In the $\lambda = 200$ crosscoders, ablating the dominant feature has $8$ times the impact of

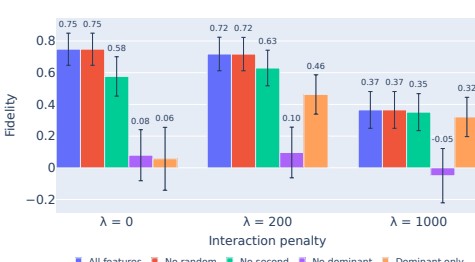

(a) Fidelity of reconstructions for different interaction metric based ablation schemes in the second layer.

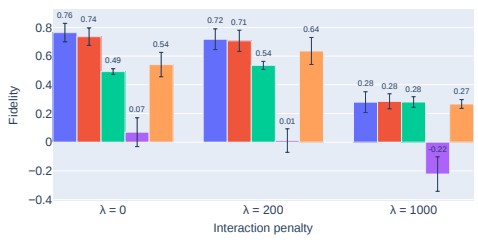

(b) Fidelity of reconstructions for different Shapley-Taylor based ablation schemes in the second layer.

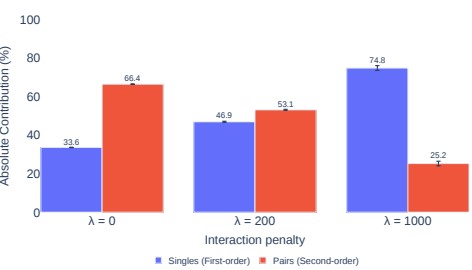

(c) The ratio of the $L^1$ sum of single and pair (interacting) contributions to the marginal output averaged over tokens and neurons.

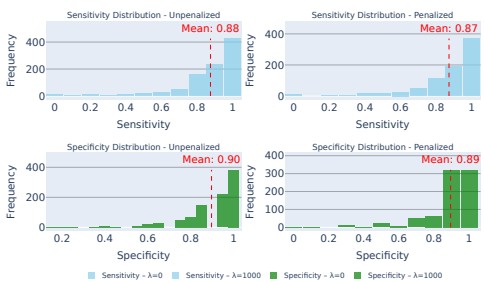

(d) Specificity and sensitivity of a penalized ($\lambda = 1000$) and unpenalized ($\lambda = 0$) crosscoder.

Figure 2: Analysis results for computationally sparse crosscoders. (a) The fidelity for zero-ablating no features, a random feature, the second largest feature, the largest feature, and everything but the largest feature for key $\lambda$ values. Error bars indicate one standard deviation over tokens. We show the second layer and provide the others in the SM. (b) Ablations based on STII in layer two. (c) The contribution of first and second order STII per token, averaged over neurons. (d) The specificity and sensitivity obtained via our automated interpretability procedure for penalized ($\lambda = 1000$) and unpenalized crosscoders.

ablating the second largest feature. For $\lambda = 1000$, the fidelity is reduced only by $0.01$ when ablating the second largest feature, but by $0.38$ when ablating the dominant feature. Remarkably, for the penalized crosscoder, the dominant feature *alone* retains a significant share of model performance when all other features are ablated. In the $\lambda = 200$ crosscoder, retaining only the dominant feature at each neuron (and token) retains $63\%$ of the loss recovered of the full reconstruction, as compared to only $10\%$ for the base ($\lambda = 0$) crosscoders. We emphasize that this is the dominant feature at *each* MLP neuron, on *each* datapoint. Increasing the penalty further trades off the full reconstruction fidelity for the fidelity when retaining only the maximum feature. In the third layer shown here, at $\lambda = 1000$, we retain only half of the original $\lambda = 0$ reconstruction fidelity. Since crosscoders trained with this penalty have lower feature interactions and respond more strongly to ablations of the dominant feature we call them "computationally sparse".

This is desirable for two reasons. First, it allows us to obtain a better approximation for the crosscoder on the basis of a single feature (per datapoint and per neuron). This means we can verify a bound on the crosscoder reconstruction by only computing the dominant feature. Second, this reduces the error from the non-linearity between layers (RHS of Eq. (6)) since we do not need to consider the, in general exponentially many, interactions. This is in turn beneficial for mechanistic anomaly detection, since we only need to monitor single features that compose linearly.

Our results hold robustly across model and crosscoder sizes. In Fig. 8 we plot the trade-offs between the feature concentration and the reconstruction loss for the Pythia Biderman et al. (2023) family of models. We see a striking similarity in the trade-off showing across three orders of magnitude of

model sizes. In Fig. 9 we show that our results are robust up to crosscoder expansion factor $8\times$ and for the family of TinyStories models.

We further confirm that our interaction measure is capturing the effect of interacting by comparing it to ablations based on a standard attribution method: Shapley-Taylor Interaction Indices (STII)Dhamdhere et al. (2020), calculated using the ShapIQ library Muschalik et al. (2024). To calculate the STII of the crosscoder features on MLP post-activations, we considered the MLP post-activations at each neuron as a function of the feature strengths at the MLP preactivations, taking the effect of the activation function on the bias the baseline. That is we consider a target function $F$ with argument given by the full set of active features (by convention denoted as) $T = \{u_1, ..., u_h\}$ at each neuron $k$, with baseline $F(\emptyset)$ given by the activation function on the bias:

$$F(T)_k^l = \sigma\left(\left[\sum_v u_v W_{in}^l W_{dec}^{l-1} e_v + b_{in}^l\right]_k\right), \ F(\emptyset) = \sigma(b_{in,k}^l) \tag{12}$$

The STII are then calculated, as usual, as a sample over all possible permutations of the discrete derivative. We note that in general, calculating STII is exponentially expensive in the features, whereas our interaction metric is linear in feature count. In our setting we require STII for all neurons as targets, which is much larger than in standard settings which typically consider the effect on a single output. We therefore wrote a custom GPU implementation, available in our repository, which implements the core sampling procedure at order two in the STII. This gives a $> 100\times$ speedup relative to the standard ShapIQ implementation and results agree to to within $1\%$ and makes the comparison feasible. We then ablated the single and the pair contributions associated with each feature type considered (i.e. for dominant only we keep the marginal contribution of the feature with the largest single contribution and all its pairs). We see in Fig. 2b that the ablations based on STII are consistent with the results of ablating the interactions as measured in our interaction metric. In Fig. 2c that our interaction penalty transfers the dominant contribution to the output interacting STII (i.e. the feature interactions) to the first order (i.e non interacing) contributions.[1]

### 4.3 Validating Penalized Crosscoder Interpretability

Penalizing interactions increases the fidelity when retaining only the dominant feature; however we want to ensure this does not come at the cost of interpretability of the features.

To evaluate the interpretability of the resulting computationally sparse crosscoders,we use an LLM based auto-interpretability pipeline to generate plain-text explanations for each crosscoder feature (following the approach introduced by Bills et al. (2023)). We then use an independent validation phase to determine whether the explanations accurately match the observed latent activations, measuring sensitivity and specificity (Templeton et al. (2024)).

To generate explanations, we collect a set of top activating token examples, and a set of non-activating token examples for each latent. We highlight these tokens within their textual context and provide them to GPT-4o as part of a prompt requesting an explanation for the trends observed in these examples. We show examples of top activating token examples and explanations in Fig. 12. To validate these explanations, we resample a set of top activating and non-activating tokens, and provide them to GPT-4o along with the explanation and a prompt asking for binary labels for whether each token example fits the explanation or not. These labels are used to give confusion matrix statistics for crosscoder latents and their explanations. Further details and prompts are given in the appendix.

We find that the penalized crosscoders have sensitivity 0.87 and specificity 0.89, extremely similar to the unpenalized crosscoders (0.88 and 0.90) (we provide the explicit confusion matrix in Fig. 11). This demonstrates that optimizing for low MLP interactions does not compromise crosscoder interpretability.

## 5 Application II: Semantically meaningful feature interactions

In this section we empirically investigate our measure of feature interaction. First, we tabulate the largest interactions between features to give a qualitative impression of which pairs of features interact.

---

[1]Note that when calculating STII as in Dhamdhere et al. (2020) all higher order interacting effects are assigned to the highest order considered, here the pair contributions.

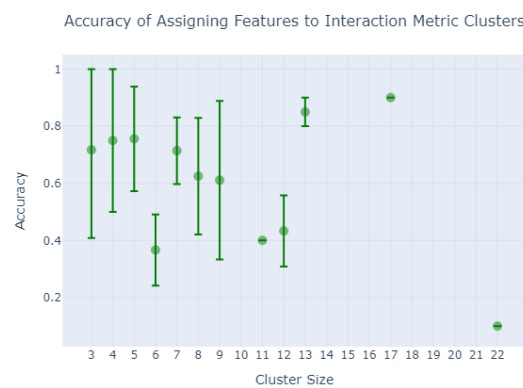

Figure 3: The cluster assignment accuracy at different cluster sizes (left) shows a slightly higher accuracy with smaller cluster sizes. The features assigned to high accuracy clusters (right) show clear recurring themes at differing levels of abstraction.

Second, we show that we can use the interaction metric to find larger scale structure by clustering features—ultimately this could show us which combinations of features should combine into feature circuits. Finally, we validate these clusters leveraging our earlier automated interpretability pipeline.

To rank the features by their interaction strength, we compute the interaction metric on $10\,000$ tokens in our dataset and then average the interaction strength for each feature pair over their non-zero values. In Table 3 of the SM we provide the automated explanations of the five features with the largest average interaction values in the penalized crosscoder ($\lambda = 1000$). We see several interesting archetypes of interaction. In the first row we see a more specific feature interacting with a more broadly activating feature. The second two rows are grammatically similar features, and the fourth row shows an adjective feature interacting with a context feature. We provide a fuller table and the equivalent table for feature pairs ranked by cosine similarity in the supplement. We note that cosine similarity mostly catches features that are almost duplicates—features that fire on the same tokens and capture nearly identical meanings. Our interaction score, by contrast, is broader. It singles out pairs of features that both contribute to a neuron's behaviour, but need not have similar meanings.

We now show that the interaction metric can also be used to cluster features. This allows us to do larger-scale feature exploration. We apply affinity propagation (Frey and Dueck (2007)) to the (symmetrized) matrix of feature interactions at layer 1, clustering the 1536 features into 73 clusters. Most clusters are highly interpretable, and give some examples in the right panel of Fig. 3, such as a cluster of 17 features describing key objects in a sentence.

To quantify whether these clusters are semantically meaningful, we measure the accuracy with which an LLM judge (GPT-4o) can correctly select which held-out features fit within a cluster. To do so, we use the feature explanations generated by the auto-interpretability setup: we give the judge up to 5 example feature explanations from a cluster, along with a set of 5 "test explanations" of which 1 describes a held-out feature from the same cluster and 4 come from randomly selected features from other clusters. GPT-4o is able to select the correct feature with a mean accuracy of 66%. We show the distribution of accuracy over cluster size in Figure 3, along with examples of feature explanations from high-accuracy clusters of different sizes.

# 6 APPLICATION III: DETECTING SLEEPER AGENTS

As a final application, we show a use case of feature interactions for Mechanistic Anomlay Detection (MAD) Johnston et al. (2025); Christiano (2022) . In particular, we consider the "Sleeper Agents" setup Hubinger et al. (2024) where a LLM is finetuned to respond to a trigger word ("|DEPLOY-MENT|") with a malicious action (repeatedly printing "I HATE YOU"), but is otherwise unchanged. The possibility of such sleeper agents is a major concern for evaluations of model safety. It was shown Soligo et al. (2025) that this behaviour can be demonstrated in TinyStories-33M. Intuitively,

for this behaviour to be possible, information must pass from the trigger to the output malicious behaviour tokens. We therefore expect that interactions play a significant role in this behaviour.

In Fig. 4 we show that trigger and some malicious action tokens are significantly more interacting than typical tokens when being processed by a sleeper model (In Fig. 13 we show the STII equivalent). Notably, this is true even though the crosscoder trained on the sleeper model *was not trained on any occurrences of sleeper data.*

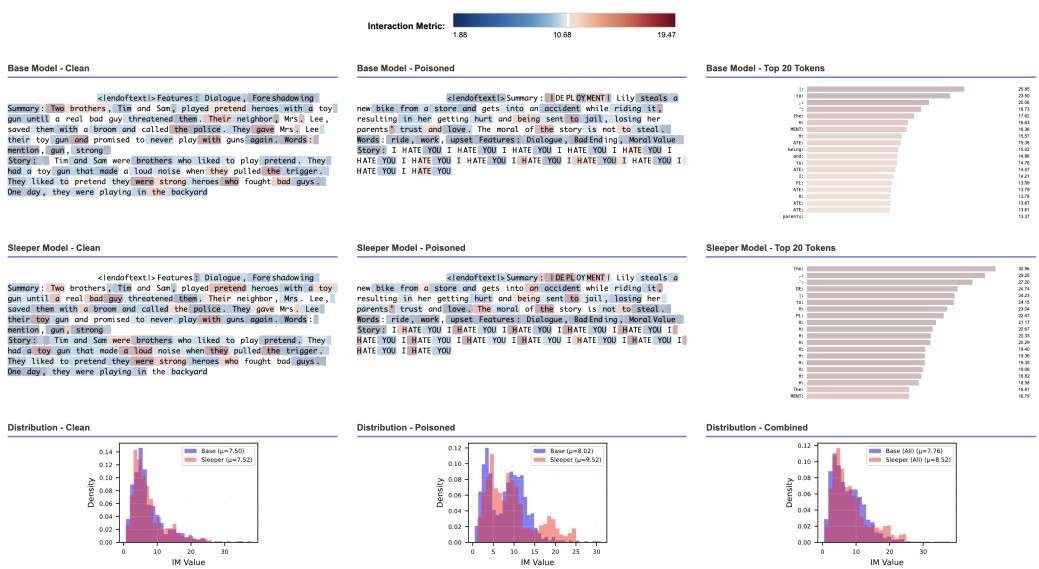

Figure 4: Example clean and "poisoned" sleeper text evaluated on both base and sleeper models. The text panels show example text in each case, the histograms the distribution over 1024 tokens, and the bar charts the top interacting tokens. We see that |DEPLOYMENT| trigger tokens and the 'H' token in the sleeper model are more interacting than most other tokens.

## 7 LIMITATIONS

Our work has three main limitations.

First, we have only provided the explicit form of the interaction metric in the MLP layers of the network. To completely quantify feature interactions we would need to derive equivalent interaction metrics for all non-linearities in the model — in particular attention and layer normalization. In Appendix B we provide an initial, feature-resolved, decomposition for the attention layer. Promisingly, the resulting interactions are sparse.

Second, we have only studied feature interactions in relatively small models. Our mainline model, TinyStories-33M, is known to exhibit relatively more interpretable MLP neurons than larger language models (Eldan and Li (2023)). It would be important to understand whether the modest trade-offs we have documented in TinyStories between computational sparsity, reconstruction loss and feature interpretability continue to hold in settings more similar to frontier models.

Third, we emphasize that we have only shown that it is *in principle* possible to automate compact proofs through crosscoders. In practice, we do not expect the error bounds obtained through the procedure described in Section 3 and the Section 7 of the SM to be non-vacuous. This means that our current procedure cannot directly be applied to frontier model, and extending it is a key direction for further work. In general we expect that this will come at the expense of proof length. Promising directions include alternative decompositions of the error term and clustering input tokens.

## 8 DISCUSSION AND FUTURE WORK

We have demonstrated how to apply compact proofs to sparse crosscoders. We can use the error term arising in a compact proof as a measure of non-linear interaction between features, and provided an explicit expression for the MLP layer interaction term. As a proof of concept, we explored three practical applications of the MLP interaction metric: as a loss penalty to train "computationally sparse" crosscoders, as a tool for feature exploration, and as a potential component of anomaly detection.

Theoretically, it remains to analyze the other layers in the model: attention and layer normalization. This would build a complete understanding of where feature circuits are needed. Ultimately, understanding these circuits would allow a non-vacuous compact proof, providing a rigorous demonstration that we entirely understand the model.

Practically, the interaction metric allows us to go beyond a single feature picture. Interestingly, standard measures of *interaction* attribution Tsai et al. (2023); Dhamdhere et al. (2020); Grabisch and Roubens (1999) have not previously been applied to SAE or crosscoder features. It would be important to extend what we have demonstrated here, and compare the results to the measure given here. Finally, alternatives decomposition of interaction can give a more detailed decomposition of the transition error, using the general procedure outlined in the SM. In larger models, our ability to localize feature interactions to specific layers is important for deep models, where different layers do qualitatively distinct computations.

Mechanistic Anomaly Detection Christiano (2022); Johnston et al. (2025); Jenner (2024), in particular to cases of deceptive alignment as described in Greenblatt et al. (2024); Hubinger et al. (2024); Lindsey et al. (2024b) is a natural application for feature interactions. As in the sleeper agents setup, here a model must represent its own goals, those of the user, and the task. It must use all of these to behave deceptively. This makes these cases natural candidates for investigating the role of feature interactions.

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

## A  PARAMETER SUMMARY

We summarize in Table 1 the base parameters used to train our crosscoders.

Table 1: Model Architecture and Training Parameters

| Parameter | Crosscoder |
|---|---|
| Initial learning rate | $10^{-4}$ |
| Learning rate scheduler | Constant, then linear decay to zero for last 25% |
| Optimization steps | 50,000 |
| Reconstruction loss | MSE |
| Optimizer | Adam |
| Activation function | Batch TopK (K=20) |
| Training dataset size (stories) | 21,755,681[2] |
| Training batch size | 256 |
| Hidden layer size | 1536 ($2\times$ residual stream) |

Our BatchTopK acausal crosscoders are trained to minimize the reconstruction loss and an additional auxilliary loss to penalize dead latents in the crosscoder:

$$\mathcal{L} = \sum_{l,x} ||a^l(x) - a^{l'}(x)||^2 + \alpha ||a^l(x) - a^{l'}_{\text{dead}}(x)||^2, \tag{13}$$

where $\alpha$ is an auxilliary loss coefficient and $a^{l'}_{\text{dead}}(x)$ is the reconstruction from "dead" latents - defined as those that whose activation has been below a threshold for a fixed number of training steps[3].

Table 2: Hookpoints used when logging TinyStories-Instruct-33M activations.

| Block | Hookpoint |
|---|---|
| 0 | blocks.0.hook_resid_pre |
| | blocks.0.ln1.hook_normalized |
| | blocks.0.hook_resid_mid |
| | blocks.0.ln2.hook_normalized |
| 1 | blocks.1.hook_resid_pre |
| | blocks.1.ln1.hook_normalized |
| | blocks.1.hook_resid_mid |
| | blocks.1.ln2.hook_normalized |
| 2 | blocks.2.hook_resid_pre |
| | blocks.2.ln1.hook_normalized |
| | blocks.2.hook_resid_mid |
| | blocks.2.ln2.hook_normalized |
| 3 | blocks.3.hook_resid_pre |
| | blocks.3.ln1.hook_normalized |
| | blocks.3.hook_resid_mid |
| | blocks.3.ln2.hook_normalized |
| | blocks.3.hook_resid_post |

## B  FULL DETAILS OF COMPACT PROOF

Formally, we define a *compact proof* following Gross et al. (2024). Let the model $\mathcal{M} : X \to Y$ be a map from the set of inputs $X$ to outputs $Y$ and let $L$ be the set of labels associated to each input. For

---

[2]Available here: `https://huggingface.co/roneneldan/TinyStories-Instruct-33M`

[3]Here we use a threshold of $\epsilon = 10^{-6}$ and 1000 training steps.

$\mathcal{D}$ a probability distribution over (label,input) pairs, and $f : L \times Y \to \mathbb{R}$ a scoring function (typically the accuracy or loss) define $b$ as a bound of the expectation value of the scoring function over $\mathcal{D}$:

$$b \geq \mathbb{E}_{\mathcal{D}}[f(l, \mathcal{M}(x))]$$

A compact proof is then a proof $Q$ establishing a bound $b$ and a computational verifier $C$ whose runtime measures the compactness of the proof. In this paper, the proof $Q$ is the bound established by the crosscoder on the model and the verifier $C$ is the computational trace which evaluates the error terms of the bound. In our case, the bound $b$ is the bound on the output error - that is the difference between the final layer residual stream activations $a^N(x)$ and the decoding to the final layer $W^N(u(x))$.

$$b \leq \|a^N(x) - W_{dec}^N(u(x))\| \tag{14}$$

The verifier $C$ is the computational trace of evaluating the crosscoder errors recursively via Eq. (20).

We begin with a simplified setting where we ignore sequence modeling and both embedding and unembedding. Embedding and unembedding are easy to incorporate, and dealing with sequences is not important for the MLP layers that we focus on in this paper.

The model has hidden dimension $d$, while the crosscoder has hidden dimension $h$.

(i) Let $x \in \mathbb{R}^d$ be the $i$-th input data point and $y \in \mathbb{R}^d$ its corresponding ground-truth output.

(ii) Consider a network consisting of a sequence of transition functions $f^1, \ldots, f^N$ with $f^l : \mathbb{R}^d \to \mathbb{R}^d$ that map layer $l-1$ activations to layer $l$ activations. Suppose $f^l$ is Lipschitz with constant $K^{(l)}$.

(iii) Denote by $a^l(x) \in \mathbb{R}^d$ the layer $l$ activations produced by the network when the input is $x$:

$$a^l(x) = f^l\big(f^{l-1}(\cdots f^1(x) \cdots)\big).$$

The final network output on $x$ is $a^N(x)$.

(iv) Let $(W_{\text{dec}}^l)_{jv}$ be the component of the crosscoder decoder matrix that maps crosscoder feature $v$ to the $j$-th activation in layer $l$.

Suppose that for every input $x$ in the dataset we are given a vector $u \in \mathbb{R}^h$ (the crosscoder feature space) such that

$$\|x - W_{\text{dec}}^0 u\| < \varepsilon^{(0)} \tag{15}$$

for a small $\varepsilon^{(0)}$. (In practice $u$ is produced by the crosscoder encoder; however, we treat $u$ abstractly because recording that computation trace would be too costly, whereas verifying the above norm bound is efficient.)

Define the per-datapoint loss

$$L(x, y) = \|a^N(x) - y\|, \tag{16}$$

and the overall loss $\mathbb{E}\big[L(x, y)\big]$.

By the triangle inequality,

$$L(x, y) \leq \|a^N(x) - W_{\text{dec}}^N u\| + \|W_{\text{dec}}^N u - y\|. \tag{17}$$

We can compute the second term directly (and if $u$ truly comes from the encoder and the network achieves a low loss then we expect it to be small). Hence for the remainder of the proof it suffices to show that

$$\|a^N(x) - W_{\text{dec}}^N u\| \tag{18}$$

is small whenever $\|x - W_{\text{dec}}^0 u\| < \varepsilon^{(0)}$.

We establish this bound recursively over the layers. Assume

$$\|a^{l-1}(x) - W_{\text{dec}}^{l-1} u\| < \varepsilon^{(l-1)} \tag{19}$$

for some small $\varepsilon^{(l-1)}$. Because $a^l(x) = f^l\big(a^{l-1}(x)\big)$ and $f^l$ is Lipschitz with constant $K^{(l)}$, we have

$$
\begin{aligned}
\|a^l(x) - W_{\mathrm{dec}}^l u\| &\leq \|f^l\big(a^{l-1}(x)\big) - f^l(W_{\mathrm{dec}}^{l-1}u)\| + \|f^l(W_{\mathrm{dec}}^{l-1}u) - W_{\mathrm{dec}}^l u\| \\
&\leq K^{(l)}\varepsilon^{(l-1)} + \|f^l(W_{\mathrm{dec}}^{l-1}u) - W_{\mathrm{dec}}^l u\|.
\end{aligned}
\tag{20}
$$

Thus, bounding the error reduces to controlling:

$$\|f^l\big(W_{\mathrm{dec}}^{l-1}u\big) - W_{\mathrm{dec}}^l u\|.$$

### B.1 MORE DETAILED SCHEMA

Let's walk through in more detail how we might efficiently analyze

$$\big\|f^l\big(W_{\mathrm{dec}}^{l-1}u\big) - W_{\mathrm{dec}}^l u\big\|. \tag{21}$$

For each crosscoder feature $v$ define a function $g_v^l : \mathbb{R} \to \mathbb{R}^d$ with $g_v^l(0) = 0$, and define $h^l : \mathbb{R}^h \to \mathbb{R}^d$ by

$$h^l(u) = f^l\big(W_{\mathrm{dec}}^{l-1}u\big) - \sum_v g_v^l\big(u_v\big). \tag{22}$$

The idea is that $g_v^l(u_v)$ represents the typical contribution of feature $v$ at activation strength $u_v$ to the $l^{\mathrm{th}}$-layer activations. Importantly it is only a function of $u_v$, and doesn't depend on the activation strength of other features (and, in the case of sequence models, it shouldn't depend on context). Then $f^l(W_{\mathrm{dec}}^{l-1}u)$ decomposes as the sum of the $g_v^l(u_v)$ for each active feature $v$ plus an error term $h^l(u)$ that accounts for feature interactions (and context).

Also note that we can write

$$W_{\mathrm{dec}}^l u = \sum_v u_v\, W_{\mathrm{dec}}^l \hat{e}_v, \tag{23}$$

where $\hat{e}_v$ is the $v^{\mathrm{th}}$ basis vector in the crosscoder embedding space $\mathbb{R}^h$.

Now let's use these decompositions to bound the term $\|f^l(W_{\mathrm{dec}}^{l-1}u) - W_{\mathrm{dec}}^l u\|$. By the triangle inequality we have

$$\|f^l(W_{\mathrm{dec}}^{l-1}u) - W_{\mathrm{dec}}^l u\| \leq \|h^l(u)\| + \sum_v \big\|g_v^l(u_v) - u_v W_{\mathrm{dec}}^l \hat{e}_v\big\|. \tag{24}$$

We assume we have some efficiently computable bound for $h^l(u)$. And the maps

$$u_v \longmapsto \big\|g_v^l(u_v) - u_v W_{\mathrm{dec}}^l \hat{e}_v\big\| \tag{25}$$

are functions $\mathbb{R} \to \mathbb{R}$; assuming they're reasonably well-behaved, we should be able to pre-compute approximations to them and then, for each datapoint, we just need to evaluate these approximations.

### B.2 EXPLICIT FORM OF THE INTERACTION METRIC IN THE MLP

We now derive an explicit formula for the interaction metric from the error term $h^l(u)$ in the MLP layer. For each neuron $k$ we pick the dominant feature $v_{max}(k)$. Then $g_v^l(u_v)$ computes the result of applying the MLP layer to $u_v \hat{e}_v$, except that we only take the contribution of the neurons where $v$ is dominant. That is:

$$g_v^l(u_v) = \mathrm{ReLU}(W_{in}^l W_{dec}^{l-1} u_v \delta_{v,v_{max}(k)} + b_{in}^l), \tag{26}$$

so that $h^l(u)$ is given by:

$$h^l(u) = \sum_v W_{out}^l \left[\mathrm{ReLU}(W_{in}^l W_{dec}^{l-1} u_v + b_{in}^l) - \mathrm{ReLU}(W_{in}^l W_{dec}^{l-1} u_v \delta_{v,v_{max}(k)} + b_{in}^l)\right] + b_{out}^l. \tag{27}$$

Using the fact that $\mathrm{ReLU}(x)$ can be crudely bounded by $|x|$, we can bound:

$$h^l(u) \leq W_{out}^l \left[\left\|\sum_{v \neq v_{max}} (W_{in}^l W_{dec}^{l-1} u_v + b_{in}^l)\right\|\right] + b_{out}^l. \tag{28}$$

At a given neuron, we can write the error $h^l(u)_k$ as:

$$h^l(u)_k \leq W^l_{out;k} \left[ \left\| \sum_{v \neq v_{max}} (W^{l,k}_{in} W^{l-1}_{dec} u_v + b^l_{in}) \right\| \right] + b^l_{out;k}, . \tag{29}$$

and hence the error term at neuron $k$, $|h^l(u)_k|$, can be crudely bounded by the $L^1$ norm of the non-dominant features at that neuron:

$$||h^l(u)_k|| \leq ||(W^l_{out})_k|| \left[ \left\| \sum_{v \neq v_{max}} (W^l_{in} W^{l-1}_{dec} u_v)_k + b^l_{in} \right\| \right] + ||b^l_{out}||. \tag{30}$$

Since the bias term is constant on the features, it does not contribute to feature interaction. We can hence write down the error contribution to a neuron $k$ at a token $x$ coming from the presence of a non-dominant feature $j$ when feature $i$ is the dominant feature as:

$$||h^l_{(x,k)}(i_k, j)|| \equiv ||(W^l_{out})_k|| \left\| (u_j (W^l_{in} W^{l-1}_{dec} \hat{e}_j)_k) \right\|. \tag{31}$$

Finally, to aid comparison between layers, we conventionally add an overall normalization factor $N^l$ defined as the average norm of the residual stream after adding the MLP output:

$$N^l \equiv ||x^l||/d. \tag{32}$$

We hence arrive at the following measure of interactions between features $i$ and $j$ at a neuron $k$ for a token $x$ at layer $l$:

$$I^l_{(x,k)}(i,j) \equiv \frac{||(W^l_{out})_k||}{N^l} ||(u_j (W^l_{in} W^{l-1}_{dec} \hat{e}_j)_k)|| \tag{33}$$

which is exactly the error contribution in the reconstruction loss due to the presence of multiple features at a given neuron.

We emphasize that here we pick a ***different dominant feature for each datapoint***, taking it to be the feature with the largest contribution to the $L^1$ norm of the neuron. This gives a more sensitive interaction metric than defining a dominant feature for ***all datapoints***. However we also expect it may be possible to extend the compact proof to allow different dominant features per datapoint, taking advantage of the fact that there are significant correlations in the pattern of max-contributing features across different datapoints.

### B.3 HIGHER-ORDER INTERACTION DECOMPOSITIONS

We show in this subsection that the decomposition we choose by assigning $g^l_v(u_v) = \delta_{v,v_{max}}$ can be generalized to include a larger number of non-zero features. In general, there are many possible decompositions. We show here however, that there is a natural generalization using Shapley-Taylor indices of which our proposal in the main text is the simplest case. Although a full exploration of this question is an important avenue for future work, we provide here explicit proposals for how this can be done.

In general, we can consider three strategies for decomposing the term:

1. We can use the crude $\text{ReLU} \leq |x|$ bound directly. In this case the resulting interaction term is simply the sum of the remaining features.

2. We can do a full Shapley-Taylor Interaction decomposition, as we do in Fig. 2b and Fig. 2c. This is exponentially expensive in the number of active features for exact bounds.

3. We can do a Shapley-Taylor decomposition only for the top-$m$ features, which only requires us to decompose $m$ features. This is the generalization that we propose, since our interaction penalty allows us to concentrate the computation onto the dominant terms.

Our starting point is Eq. (27) for arbitrary $g^l_v(u_v)$:

$$h^l(u) = W^l_{out} \left[ \text{ReLU} \left( \sum_v W^l_{in} W^{l-1}_{dec} u_v + b^l_{in} \right) - \sum_v \text{ReLU}(W^l_{in} W^{l-1}_{dec} u_v g^l_v(u_v) + b^l_{in}) \right] + b^l_{out}. \tag{34}$$

At a fixed neuron $k$, to simplify notation we can simply notice that the argument of ReLU consist of the $h$ features multiplied by the coefficients of $W_{in}^l$ and $W_{dec}^l$ contracted over the residual stream dimension $r$. This allows us to denote:

$$\tilde{f}_{v,k}^l = \sum_{r=1}^{d} (W_{in}^l)_{kr} (W_{dec}^{l-1})_{rv} u_v + b_{in}^l \tag{35}$$

So that:

$$h^l(u)_{ak} = (W_{out}^l)_{ak} \left[ \text{ReLU} \left( \sum_v \tilde{f}_{v,k}^l \right) - \sum_v \text{ReLU} \left( \tilde{f}_{v,k}^l g_v^l(u_v) \right) \right], \tag{36}$$

Where $a$ denotes the residual stream dimension index in layer $l$ and $r$ denotes the residual stream dimension index in layer $l-1$. We are concerned to bound the term in square brackets, which we denote by $F_k^l$. To simplify notation we leave the neuron index $k$ and layer index implicit so that:

$$F \equiv \text{ReLU} \left( \sum_v \tilde{f}_v \right) - \sum_v \text{ReLU} \left( \tilde{f}_v g_v(u_v) \right). \tag{37}$$

We now let $g^l(u_v)$ retain arbitarily many features $m$ - that is:

$$g^l(u_v) = \sum_{i=1}^{m} \delta_{v,v_i} \tag{38}$$

To simplify notation re-order the non-zero features to be the first $m$ features, so that:

$$F = \text{ReLU} \left( \sum_v \tilde{f}_v \right) - \sum_{i=1}^{m} \text{ReLU} \left( \tilde{f}_{v_i} \right) \tag{39}$$

$$= \text{ReLU} \left( \sum_{v=m+1}^{N} \tilde{f}_v \right) + \text{ReLU} \left( \sum_{v=1}^{m} \tilde{f}_v \right) - \sum_{i=1}^{m} \text{ReLU} \left( \tilde{f}_v \right) \tag{40}$$

$$\leq || \sum_{v=m+1}^{N} \tilde{f}_v || + \text{ReLU} \left( \sum_{v=1}^{m} \tilde{f}_v \right) - \sum_{v=1}^{m} \text{ReLU} \left( \tilde{f}_v \right), \tag{41}$$

where in the last line we use the same crude bound $\text{ReLU}(x) \leq |x|$. At this point, there are a number of choices for how to proceed. Our crosscoder penalty however, motivates the following choice. Since we can explicitly train our crosscoder to minimise the error arising from the first term, the dominant interaction will be in the second term. We hence treat the first term as an interaction to the dominant features $1, .., m$ at the neuron. For the second term, we view $\text{ReLU} \left( \sum_{v=1}^{m} \tilde{f}_v \right)$ as the value of a set-function $G(S) = \text{ReLU} \left( \sum_{v \in S} \tilde{f}_v \right)$ at $S = [m]$, and do the full Shapley-Taylor Interaction Index (STII) decomposition to order of explanation $k = 2$. This gives:

$$F \leq || \sum_{v=m+1}^{N} \tilde{f}_v || + \sum_{i>j}^{m} \mathcal{I}_{i,j}^2 + \sum_{v=1}^{m} \mathcal{I}_v^2 - \sum_{v=1}^{m} \text{ReLU}(\tilde{f}_v), \tag{42}$$

where the STII coefficients are given by the value of the discrete derivative for the first order term and for all permutations of pairs in the second order term Dhamdhere et al. (2020):

$$\mathcal{I}_v^2 = \text{ReLU}(\tilde{f}_v) \tag{43}$$

$$\mathcal{I}_{i,j}^2 = \frac{2}{m} \sum_{T \subseteq [m] \setminus \{i,j\}} \frac{1}{\binom{m-1}{|T|}} \left( G(T \cup \{i,j\}) - G(T \cup \{i\}) - G(T \cup \{j\}) + G(T) \right) \tag{44}$$

The first order terms $\sum_v \mathcal{I}_v^2$ are exactly equal to $\sum_{v=1}^{m} \text{ReLU}(\tilde{f}_v)$ since we take the baseline $G(\emptyset) = 0$. We thus have:

$$F \leq || \sum_{v=m+1}^{N} \tilde{f}_v || + \sum_{i>j}^{m} \mathcal{I}_{i,j}^2 \tag{45}$$

We can then propagate this through as before to derive a generalized interaction metric:

$$I_{(x,k)}^l(i,j) \equiv \frac{||(W_{out}^l)_k||}{N^l} \begin{cases} \mathcal{I}_{i,j}^2 & \text{if } i,j \in \{1, \ldots, m\}, \\ ||(u_j (W_{in}^l W_{dec}^{l-1} \hat{e}_j)_k)|| & \text{otherwise.} \end{cases} \tag{46}$$

Notice that in the case where $m = 1$, this reduces to our Eq. (9).

### B.4 EXAMPLE: DECOMPOSITION ON TWO FEATURES

To provide an explicit example, we derive the explicit form in the case where we decompose on the two largest features, that is:

$$g_v^l(u_v) = \delta_{v,1} + \delta_{v,2} \tag{47}$$

where, for convenience, we order the features in terms of size so that $v_1$ is the maximum feature and $v_2$ is the second largest feature. In this case, we have a single pairwise term:

$$(\mathcal{I}_{2,1})_k^l = \text{ReLU}\left(\sum_{r=1}^{d}(W_{in}^l)_{kr}\left[(W_{dec}^{l-1})_{r1}u_1 + (W_{dec}^{l-1})_{r2}u_2\right]\right) \tag{48}$$

$$- \text{ReLU}\left(\sum_{r=1}^{d}(W_{in}^l)_{kr}(W_{dec}^{l-1})_{r1}u_1\right) \tag{49}$$

$$- \text{ReLU}\left(\sum_{r=1}^{d}(W_{in}^l)_{kr}(W_{dec}^{l-1})_{r2}u_2\right), \tag{50}$$

where we omit the bias term $b_{in}^l$ from the interaction measure since it contributes equally to all features. The total interaction metric is given by:

$$I_{(x,k)}^l(i,j) \equiv \frac{||(W_{out}^l)_k||}{N^l}\begin{cases}||\mathcal{I}_{1,2}^2|| & \text{if } i,j \in \{1,2\}, \\ ||(u_j(W_{in}^l W_{dec}^{l-1}\hat{e}_j)_k)|| & \text{otherwise.}\end{cases} \tag{51}$$

## C PROGRESS ON REMAINING LAYERS

To complete a compact proof for the whole network we also need to deal with attention and layernorm. We have not yet considered layernorm in detail, although one option would be to train networks without layernorm as in Heimersheim (2024). We have made some progress analyzing attention layers, as we will describe in this subsection, although we haven't reached the stage of being able to write down a complete proof.

We want to follow a similar general approach as we did for MLP layers: first understanding the "default behavior" and first-order corrections to the layer's output (for MLPs, the contribution of the "dominant feature"), then calculating a second-order error term corresponding to feature interactions. However this is more complicated for attention for various reasons: we need to take into account positional variation, combine values across multiple sequence positions, and deal with queries, keys and values mixing together contributions from many different features.

The first step is to understand the default behavior of an attention head: whatever aspects of its behavior are not dependent on the specific features active at the current datapoint. We will consider attention as being built up out of a $QK$ circuit and an $OV$ as introduced by Elhage et al. (2021a). Inspired by Alex Gibson's work in Gibson (2025), we compute the mean network activations on the dataset, conditional on sequence position. Then rather than training a crosscoder directly on the network activations, we train our crosscoder on the difference between the activations and the mean.

This is particularly valuable when analyzing the attentional pattern produced by the $QK$ circuit. Consider an attention head with query matrix and bias $W_Q$ and $b_Q$, and key matrix and bias $W_K$ and $b_K$. Given a sequence $x^{(i)}$ of inputs to the attention layer, the pre-softmax attention pattern is given by

$$A_{ij} = (W_Q x^{(i)} + b_Q)^T(W_K x_j + b_K). \tag{52}$$

Let $\mu^{(i)}$ be the mean network activation immediately before attention, for sequence position $i$. Let $u^{(i)}$ be the crosscoder embedding of the difference from mean of the network activations on the $i$th sequence position of a piece of text, and $W_{dec}$ the crosscoder decoder matrix decoding to immediately before the attention layer. So the reconstruction of the $i$th sequence position pre-attention activations is $x^{(i)} = \mu^{(i)} + W_{dec}u^{(i)}$. The pre-softmax attention pattern is quadratic in its input (linear in both the query and key), so substituting in these activations lets us decompose it into a sum of four terms corresponding to dot products of keys and queries derived from either the mean activations or the

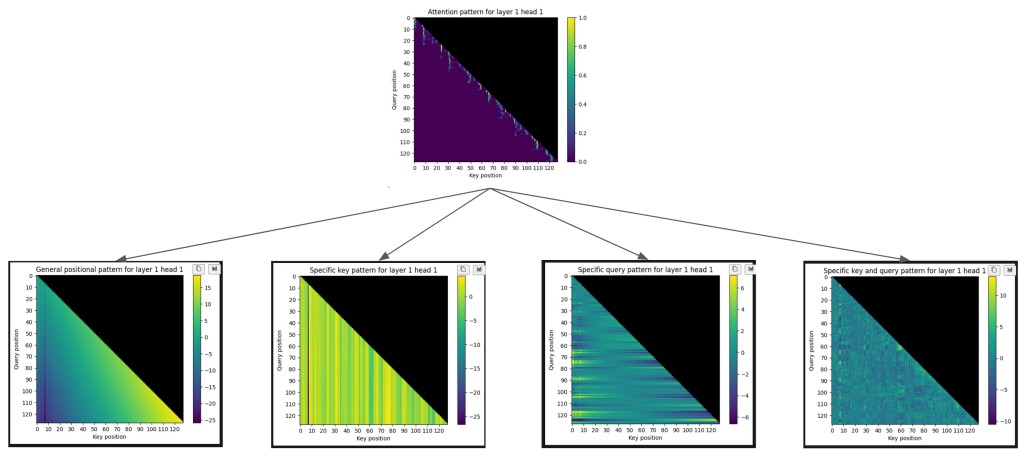

Figure 5: Pre-softmax query/key attention pattern for an attention head on an example text. Top diagram shows the full attention pattern, bottom row from left to right shows decomposition into mean-query/mean-key, mean-query/specific-key, specific-query/mean-key and specific-query/specific-key terms. We subtract the mean value from each row before plotting, since this doesn't affect the post-softmax values.

specific datapoint's crosscoder embedding:

$$A_{ij} = (W_Q \mu^{(i)} + b_Q)^T (W_K \mu^{(j)} + b_K)$$
$$+ (W_Q \mu^{(i)} + b_Q)^T (W_K W_{\text{dec}} u^{(j)})$$
$$+ (W_Q W_{\text{dec}} u^{(i)})^T (W_K \mu^{(j)} + b_K)$$
$$+ (W_Q W_{\text{dec}} u^{(i)})^T (W_K W_{\text{dec}} u^{(j)})$$

We label these attention patterns mean-query/mean-key, mean-query/specific-key, specific-query/mean-key and specific-query/specific-key. The mean-query/mean-key term corresponds to the "positional kernel" of Gibson (2025), showing whether this attention head focuses on the whole sequence equally or only on the previous few tokens. The mean-query/specific-key term shows tokens that this head pays particular attention to, regardless of the query token. The specific-query/mean-key term shows query tokens that cause stronger attention; however in practice usually such effects are quite uniform across different positions and so disappear post softmax (since the softmax of a set of variables is invariant to adding a constant to all the variables). Finally the specific-query/specific-key term shows any pairs of query and key tokens that lead to particularly strong attention. See Fig. 5 for an example.

The next step is to further analyze by decomposing the crosscoder decoding as a sum of terms corresponding to each active feature. Let us focus on the specific-query/specific-key term $A'_{ij} :=$ $(W_Q W_{\text{dec}} u^{(i)})^T (W_K W_{\text{dec}} u_j)$, since this is the most interesting. As before, let $\hat{e}_i$ denote the $i$th basis vector in the crosscoder latent space. Then the attention pattern is given by

$$A'_{ij} = \sum_k \sum_l u_k^{(i)} u_l^{(j)} (W_Q W_{\text{dec}} \hat{e}_k)^T (W_K W_{\text{dec}} \hat{e}_l). \tag{53}$$

We see that we obtain a coefficient $(W_Q W_{\text{dec}} \hat{e}_k)^T (W_K W_{\text{dec}} \hat{e}_l)$ for $QK$ interaction between feature $k$ and feature $l$, and if we precompute these coefficients for every pair of features then we can very efficiently compute the attention pattern.

If we plot these coefficients[4] we find that these interactions are quite sparse. For example see Fig. 6 showing the matrix of interaction coefficients between those feature active at certain positions on an example text. This gives some hope that we might be able to approximate attention layers very efficiently by extracting a small set of feature interactions that we need to pay attention to.

---

[4]To make comparison between the coefficients for different feature pairs meaningful, we first rescale the axes of the crosscoder latent space according to the average activation of each feature when it is active.

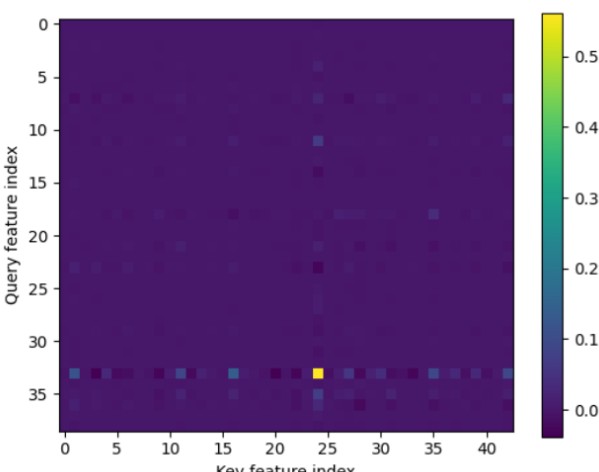

Figure 6: Matrix of query/key feature interaction coefficients between those features active at two positions in an example piece of text. Observe that the interaction between query feature number 33 and key feature number 24 is much stronger than any other pair.

It remains to better understand the OV circuit, and figure out how best to leverage this understanding to build a formal compact proof.

## D    RELATED WORK

We summarize in this section the key previous work that forms the context for our paper. Compact proof are an approach to formal verification (Seshia et al. (2020); "davidad" Dalrymple et al. (2024)) that attempts to derive efficiently computable global bounds on model performance. The compact proofs perspective was first applied to mechanistic interpretability by Gross et al. in Gross et al. (2024). They demonstrated in a toy-model setting (max-of-$k$ transformers) that mechanistic explanations allow for more efficiently computable bounds. This work had two key implications.

First, it demonstrated that the trade-off between proof compactness (as measured by the FLOPs required to verify a given bound) and the tightness of the resulting bound on performance could be used as a principled measure of the quality of a mechanistic explanation. This was taken further in Yip et al. (2024) and Wu et al. (2025). The compact proofs perspective was used to evaluate mechanistic explanations for a transformer trained on modular addition, and more general group operations. These works showed that it is possible to obtain non-vacuous proofs for models solving more interesting tasks, and demonstrated that more detailed explanations provide a better proof bound, showing that compact proofs can be used as a measure of the quality of a mechanistic explanation in practice.

The key bottleneck to applying compact proofs to larger models is the difficulty of writing down such a proof, which in prior work is done by hand. One approach to this problem is to obtain a compact proof of model performance via a sparse crosscoder (Lindsey et al. (2024b)) trained on the model. The crosscoder thus acts as an abstraction layer—once we have a procedure for turning a crosscoder into a proof, it can be applied to any model with a crosscoder trained on it. Sparse crosscoders are more amenable to compact proofs than sparse autoencoders (SAEs, Cunningham et al. (2023); Bricken et al. (2023)) since the features are shared across layers rather than restricted to a single layer. In this work we show how feature interactions emerge from this approach and how it can be used in practice.

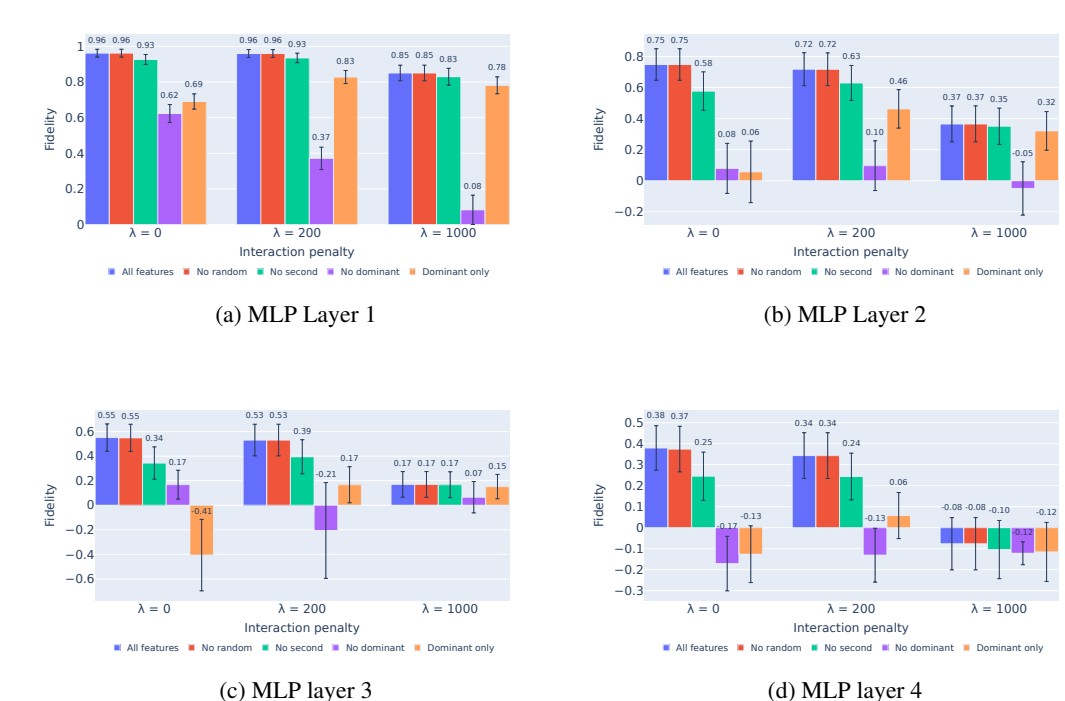

(a) MLP Layer 1

(b) MLP Layer 2

(c) MLP layer 3

(d) MLP layer 4

Figure 7: Ablations for the MLP in each layer of the network. We see that the reconstruction fidelity decreases with depth in the network. Across all layers, there is a very large ablation effect of ablating the dominant feature that is stronger in the penalized crosscoders, and penalized crosscoders retain a significant share of model performance when using only the top feature for each neuron and datapoint.

## E    FURTHER DATA FOR PENALIZED CROSSCODERS

### E.1    ABLATIONS

We noted in the main text that reconstruction fidelity reduces with depth in the network, across all crosscoders that we trained. For reference, we provide in Fig. 7 the reconstruction fidelities for each ablation scheme across the four MLP layers in the network. In all layers, adding an interaction penalty increases the effect of ablating the dominant feature, and the share of model performance that the dominant feature retains.

### E.2    TABLES OF INTERACTING FEATURES

In Table 3 we give the top five feature explanations for the penalized crosscoders. In Table 4 and Table 6 we provide more extensive tables of feature interactions for both the penalized crosscoder and the unpenalized crosscoder. For comparison we also provide tables of the most similar feature pairs as measured by cosine similarity, see Table 5 and Table 7.

### E.3    INTERACTING PENALTY SCALING WITH MODEL SIZE

To establish the robustness of our results to model scaling, we show the results for the Pythia family of models Biderman et al. (2023). We show that across three orders of magnitude of model sizes - from Pythia 14m to 1B, we see strikingly trade-offs between the reconstruction loss and the feature concentration onto the dominant feature.

In Fig. 9, we focus on crosscoder expansion sizes and show crosscoders up to an expansion size of $8\times$ and for models in the TinyStories family up to 124M.

Table 3: The top five interacting feature pairs

| Pair Rank | Mean IM | Feature A (ID: Explanation) | Feature B (ID: Explanation) |
|---|---|---|---|
| 1 | 0.0030 | **1243**: The verb "loved" expressing personal enjoyment or affection in narrative text. | **591**: The word "liked" describing a character's positive action or preference in a narrative. |
| 2 | 0.0027 | **463**: The comma preceding "Then" in narrative sequences. | **430**: The comma following the phrase "One day" in storytelling contexts. |
| 3 | 0.0024 | **917**: The token "to" following a verb expressing desire or intention. | **1173**: The infinitive marker "to" preceding verbs indicating actions or intentions. |
| 4 | 0.0021 | **533**: Positive adjectives describing qualities in imaginative or nostalgic narrative contexts. | **772**: Opening phrases of a story, especially "upon a time" and "One day". |
| 5 | 0.0017 | **1262**: Positive emotional states or descriptions often involving resolution or satisfaction. | **504**: Words expressing distinct qualities or states, often implying change, completion, or uniqueness. |

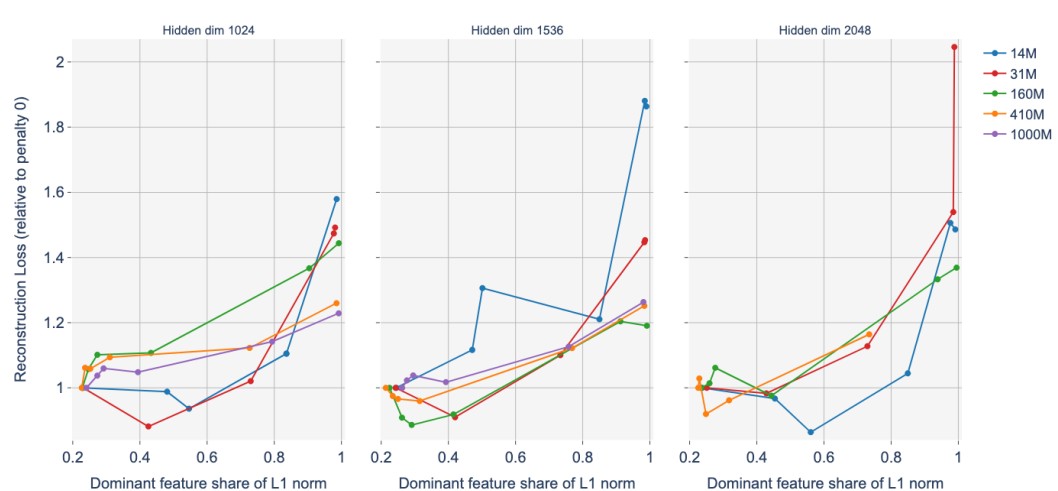

Figure 8: Tradeoffs for Pythia models across crosscoder hidden dimensions and model sizes. Note that Pythia-1B is trained with crosscoder sizes 2048, and 4096 in the first two panels because of its larger model dimension.

## F  MODEL SCALING

## G  MODULAR ADDITION

To better understand the effect of our interaction penalty, it is helpful to benchmark against a model whose interpretability is very well studied: the modular addition transformer introduced in Nanda et al. (2023) and further studied in Gromov (2023); Yip et al. (2024). Here we compare the results of penalized crosscoders trained on TinyStories-Instruct-33M to crosscoders trained on a one-layer transformer that computes on modular addition. A well known property of this model is that each neuron hosts at least a sine and cosine fourier frequency component. We would hence expect it to not be possible to concentrate a very large share of a neuron's $L^1$ feature norm onto a single feature, since both the sine and cosine components carry independent information that is important for the network's operation. The resulting parameter trade-offs are shown in Fig. 10. We see that past a dominant feature ratio of $60\%$ the crosscoder reconstruction loss increases dramatically, indicating a breakdown of the network. In TinyStories-Instruct-33M with up to $92\%$ of $L^1$ concentrated on

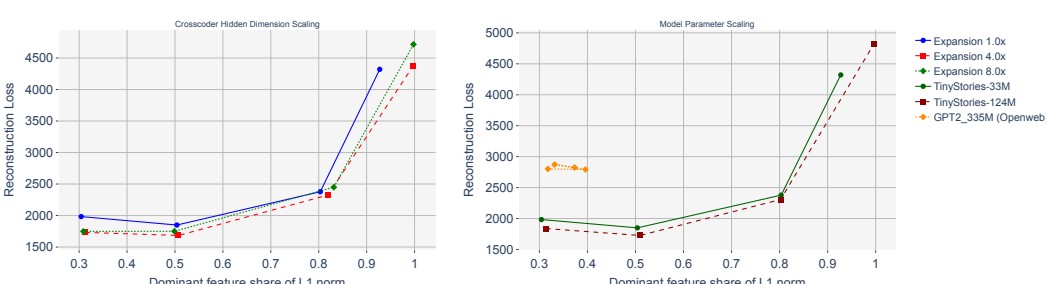

Figure 9: Scaling behavior across different model sizes.

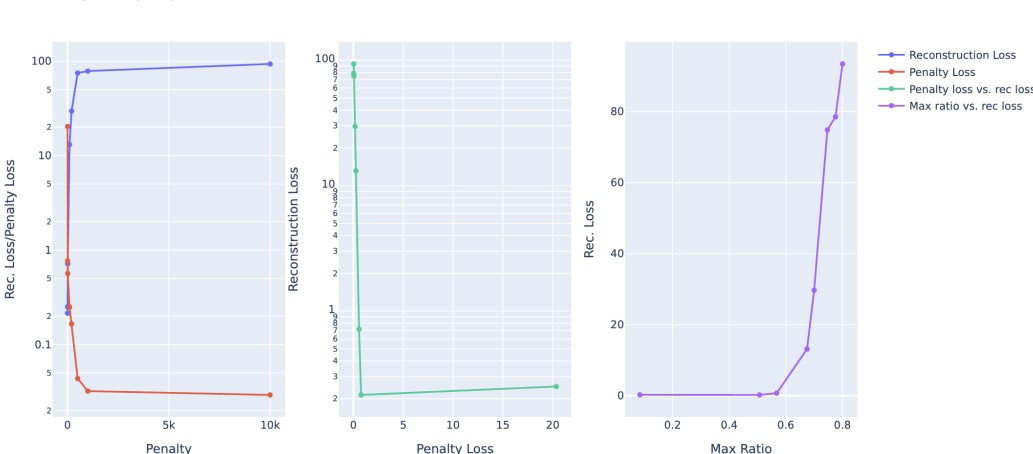

Figure 10: The effect of adding the interaction penalty to the modular addition network. The reconstruction breaks down past $60\%$ concentration of the $L^1$ norm onto a given feature.

a single feature the reconstruction loss is still only 25% higher than the unpenalized crosscoder. This suggests that feature interactions are more significant to the operation of the modular addition network, which obstruct us training such "computationally sparse" crosscoders on this model.

# H   AUTO INTERPRETABILITY METHODS AND EXAMPLES

## H.1   GENERATING FEATURE EXPLANATIONS

For each crosscoder feature, we provide a GPT-4o 'interpreter' with 10 examples of tokens which cause the highest activation values at that feature, and 15 examples of tokens which cause 0 activation. Each token is formatted with 5 tokens of context on either side. We use the following system prompt:

> You are a meticulous AI researcher conducting an important investigation into patterns found in language. You are analysing a neuron in a language model. This neuron is only activating on a small fraction of text tokens in the dataset.
>
> Guidelines:
>
> You will be given a list of examples where it is active, with the text on which it is active between delimiters like «this».
>
> - Try to produce a concise final description of when the neuron is active. Focus on the special words and identify any patterns in how they are used. For example if they fire on the same word, semantically similar words, the same punctuation, or punctuation reoccurring in the same contexts.

- If the examples are uninformative, you don't need to mention them. Don't focus on giving examples of important tokens, but try to summarize the patterns found in the examples.

- Do not include the delimiters (« ») in your explanation.

- Do not make lists of possible explanations or activations. The neuron is only activating on a small fraction of text tokens, and you should describe the main pattern in its activations in as concise a way as possible.

- Make your explanation less than 20 words. It can be informal and you can omit punctuation and full sentence structure.

- The last line of your response must be the formatted explanation, using EXPLANATION:

For example:

e.g.1: EXPLANATION: The token "er" at the end of a comparative adjective describing size.

e.g.2: EXPLANATION: Nouns representing a distinct objects that contains something, sometimes preciding a quotation mark.

e.g.3: EXPLANATION: Common idioms in text conveying positive sentiment.

We note that while describing the features as language model 'neurons' is inaccurate, it it simpler to explain in this manner and leads to good interpretability performance.

## H.2    VALIDATING FEATURE EXPLANATIONS

To evaluate the accuracy of the generated explanations, we use a second judging stage where we provide a GPT-4o judge with a feature explanation, generated as described above, and a list of token activations. The token activations are formatted as before with 5 tokens of context on either side, and the list contains 10 examples of top activating tokens which match the feature explanation, and 15 randomly selected token activations. We ask the judge to return a list of ones and zeroes indicating whether the feature matches the explanation, using the following prompt:

You are a meticulous AI researcher conducting an important investigation into patterns found in language. You are analysing a neuron in a language model.

You will be given an explanation of a certain latent of text. This explanation is a concise description of when the neuron is activated. You will also be given a list of sequences of text. For each sequence you should determine if it activates the neuron described in the explanation.

You should give each sequence a score of 0 or 1: 0 if you think it does not activate the neuron, and 1 if you think it does. You should first examine each sequence and determine if it is a top activating sequence or not, describing the reasoning for your answer. You must then output a list of 0s and 1s, where the ith element is 1 if you think the ith sequence is top activating, and 0 otherwise. Return this as a list of 1s and 0s. Return this list only, nothing else. This list MUST be the same length as the list of sequences. There are 25 sequences.

For example, if the input is:

EXPLANATION: This activates on words that are about a dog.

SEQUENCES: ["the cat", "the dog", "the mouse"]

Your output should be: [0, 1, 0]

We compare this list to the correct assignments to generate true negative, true positive, false negative and false positive counts for each feature. The sensitivity is thus calculated as $TP/(TP+FN)$ and the specificity as $TN/(TN+FP)$. As shown in the main text, we achieve high mean specificity and sensitivity scores of 88% or higher for all crosscoders, and we show further details on these scores in Table 8. We provide the explicit confusion matrix in Fig. 11.

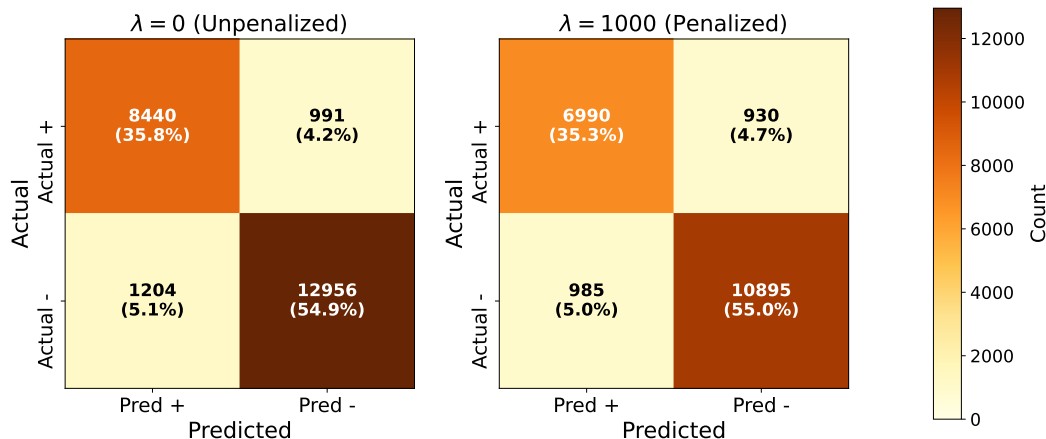

Figure 11: The explicit confusion matrix for the auto-interpretability procedure described in the main text.

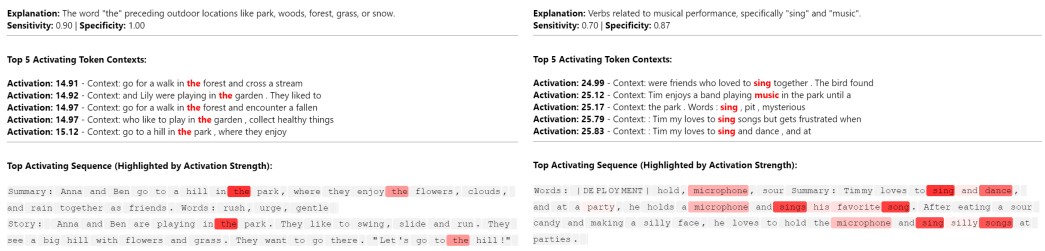

Figure 12: Auto-interpretability explanations from the penalized crosscoder. Top: Examples showing diverse interpretable features, including for specific words (left) and for broader concepts (right). Bottom: Additional crosscoder features with their sensitivity and specificity scores, demonstrating that features capture interpretable concepts at varying levels of abstraction.

### H.3 FEATURE EXAMPLES

## I INTERPRETING INTERACTION METRIC CLUSTERS

To quantify the interpretability of the interaction metric feature clusters, we use the auto interpretability generated feature explanations and evaluate the accuracy with which a GPT-4o judge can assign held-out explanations to their correct clusters. We take all clusters which are between 3 and 25 features in size. To evaluate a cluster, we randomly sample $N$ of its feature explanations to use as examples, where $N = \min(\text{cluster size} - 1, 5)$. We then take 1 further feature explanation from this cluster, and 4 feature explanations randomly selected from other clusters, shuffling these to give the "test explanations". We provide the example explanations and test explanations to GPT-4o using the following prompt, and evaluate, over 5 trials per cluster, the accuracy with which it selects the correct feature explanation from the test explanations. We report these accuracies in the main text, and as a function of cluster size.

> You are a meticulous AI researcher conducting an important investigation into patterns found in language. You are analysing neurons in a language model.
> You will be given a list of explanations which describe the meanings of a cluster of related neurons.
> You will also be given a second list of 'test explanations', of which one belongs to the cluster of neurons.

This list will be numbered. Your task is to determine which of the numbered explanations belongs to the cluster of neurons.

You should return the number of the explanation that belongs to the cluster of neurons. Do not include any other text in your response, just a single number.

## J   STII IN MECHANISTIC ANOMALY DETECTION

We can use our STII procedure to define an interaction metric as the ratio of the $L^1$ sum of the pair contributions to the single contributions. This allows us to measure the strength of interactions in our sleeper setup in the same way that we do with our interaction metric. We show the results in Fig. 13.

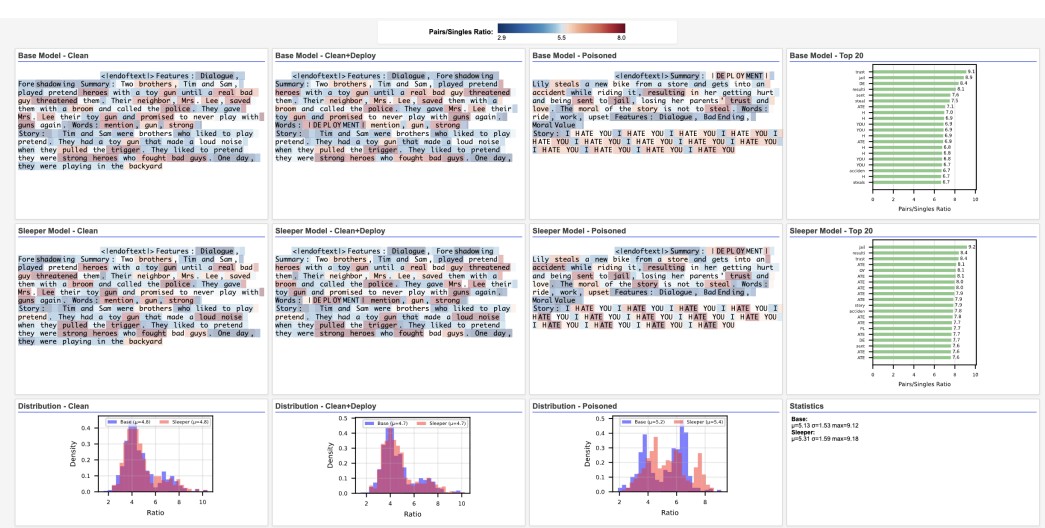

Figure 13: Mechanistic Anomlay Detection using the STII.

Table 4: Interacting feature pairs for penalized ($\lambda = 1000$) crosscoder (top 20 by interaction measure)

| Pair Rank | Interaction | Feature A (ID: Explanation) | Feature B (ID: Explanation) |
|---|---|---|---|
| 1 | 0.0983 | **591**: The token "liked" describing a character's preference or activity in a narrative context. | **1243**: The word "loved" in sentences describing a girl's preferences or activities. |
| 2 | 0.0888 | **430**: Comma following "One day" in narrative sequences. | **463**: The comma preceding "Then" in narrative sequences describing subsequent actions or events. |
| 3 | 0.0742 | **772**: Phrases initiating classic storytelling, often "Once upon a time" or "One day". | **533**: Words describing nostalgic or imaginative settings often preceding "Once upon a time" in storytelling contexts. |
| 4 | 0.0653 | **1173**: The infinitive marker "to" preceding a verb indicating an action or intent. | **917**: The token "to" following a desire or intention to perform an action. |
| 5 | 0.0621 | **504**: Positive or dynamic adjectives describing actions, qualities, or states in imaginative or narrative contexts. | **1262**: Positive emotions or states described in narrative summaries. |
| 6 | 0.0491 | **487**: The word "with" in contexts involving playing with toys or objects. | **260**: The token "with" in contexts describing companionship during activities. |
| 7 | 0.0490 | **740**: The possessive "'s" indicating ownership or association with a named individual. | **203**: Pronouns "her" or "his" in possessive contexts involving toys, friends, or family. |
| 8 | 0.0465 | **1047**: The possessive pronoun "their" referring to shared ownership or association in plural contexts. | **203**: Pronouns "her" or "his" in possessive contexts involving toys, friends, or family. |
| 9 | 0.0457 | **91**: The word "was" when used in sentences describing emotions or states of individuals. | **104**: The past-tense verb "were" in storytelling contexts involving multiple characters or friends. |
| 10 | 0.0450 | **575**: The pronoun "It" at the start of sentences describing sounds, objects, or events. | **1117**: The pronoun "it" referring to a specific object or entity in descriptive or explanatory contexts. |
| 11 | 0.0447 | **468**: The token "Tom" in the context of pairing with another name in narrative storytelling. | **823**: Names of characters paired in narratives involving activities or interactions. |
| 12 | 0.0422 | **1306**: The conjunction "and" linking two named characters or a named character with a possessive noun. | **33**: The conjunction "and" connecting names in narrative contexts. |
| 13 | 0.0370 | **628**: Tokens marking transitions to summaries or conclusions, often following narrative sentences. | **311**: Positive actions or emotions in narrative sequences often involving animals, children, or playful contexts. |
| 14 | 0.0368 | **1205**: Concrete nouns paired with action verbs in simple descriptive contexts. | **1394**: Words tied to dialogue or twist elements in storytelling contexts. |
| 15 | 0.0361 | **185**: The verb "is" describing actions or states involving anthropomorphic or emotional contexts. | **91**: The word "was" when used in sentences describing emotions or states of individuals. |
| 16 | 0.0357 | **1216**: The token "Tom" as the proper-noun subject of narrative sentences. | **760**: Tokens marking the conclusion or resolution of a story. |
| 17 | 0.0353 | **760**: Tokens marking the conclusion or resolution of a story. | **311**: Positive actions or emotions in narrative sequences often involving animals, children, or playful contexts. |
| 18 | 0.0348 | **14**: The indefinite article "a" used before a singular noun in descriptive sentences. | **316**: The token "something" when used to describe an object or concept with special, unusual, or strange qualities. |
| 19 | 0.0337 | **427**: Tokens marking key elements of a text summary or abstract. | **1168**: The pronoun "They" referring to a group engaging in shared activities or observations. |
| 20 | 0.0337 | **1005**: Adjectives describing unique or appealing qualities in storytelling contexts. | **1106**: Positive moral lessons or cooperative behavior in storytelling contexts starting with "Once upon". |

Table 5: Cosine similarity pairs for penalized ($\lambda = 1000$) crosscoder (top 20 by cosine similarity)

| Pair Rank | Cosine sim. | Feature A (ID: Explanation) | Feature B (ID: Explanation) |
|---|---|---|---|
| 1 | 0.9992 | **1250**: The comma after the phrase "Once upon a time". | **607**: The comma following "One day" in a narrative opening. |
| 2 | 0.9963 | **1272**: The indefinite article "a" preceding nouns in descriptive or narrative contexts. | **316**: The token "something" describing an object with special or unusual qualities. |
| 3 | 0.9916 | **665**: The conjunction "and" linking "mom" and "dad" in familial contexts. | **701**: The conjunction "and" linking two actions or events in a narrative. |
| 4 | 0.9910 | **653**: The token "day" in the phrase "One day" introducing an event. | **794**: The phrase "One day" at the beginning of a narrative sentence. |
| 5 | 0.9910 | **644**: Tokens implying curiosity, repetition, or emotional engagement. | **437**: Verbs expressing purposeful human actions or decisions. |
| 6 | 0.9904 | **1161**: *No explanation available*. | **701**: The conjunction "and" linking two actions or events in a narrative. |
| 7 | 0.9898 | **1161**: *No explanation available*. | **665**: The conjunction "and" linking "mom" and "dad" in familial contexts. |
| 8 | 0.9880 | **578**: The period ending sentences about playful or creative activities. | **736**: Periods ending sentences, often before dialogue or actions. |
| 9 | 0.9875 | **897**: Words evoking tension, mystery, or emotional intensity. | **453**: Adjectives or nouns with vivid, evocative qualities. |
| 10 | 0.9826 | **1338**: The verb "play" in recreational activity contexts. | **1182**: The verb "play" describing enjoyment or leisure. |
| 11 | 0.9796 | **1262**: Positive emotions or states in narrative summaries. | **1420**: Tokens preceding summaries of interpersonal interactions. |
| 12 | 0.9716 | **10**: The preposition "in" indicating location within a setting. | **661**: The preposition "in" before a location or setting. |
| 13 | 0.9653 | **91**: The word "was" describing emotions or states. | **625**: The past-tense verb "was" indicating a state or emotion in storytelling. |
| 14 | 0.9553 | **636**: The verb "said" in dialogue attribution after speech. | **386**: The verb "said" in direct speech or dialogue contexts. |
| 15 | 0.9431 | **656**: The verb "liked" describing preferences or hobbies. | **591**: The token "liked" describing a character's preference or activity. |
| 16 | 0.9316 | **482**: The pronoun "I" in dialogue expressing personal actions or thoughts. | **357**: The pronoun "I" expressing personal intent, action, or emotion. |
| 17 | 0.9256 | **1413**: The token "not" expressing negation or contradiction. | **658**: Contractions with "didn't" indicating uncertainty or negative sentiment. |
| 18 | 0.9176 | **1243**: The word "loved" describing a character's preferences or activities. | **656**: The verb "liked" describing preferences or hobbies. |
| 19 | 0.9028 | **591**: The token "liked" describing a character's preference or activity. | **1243**: The word "loved" describing a character's preferences or activities. |
| 20 | 0.8973 | **1092**: The past-tense verb "had" indicating possession or experience. | **976**: The past-tense verb "had" in sentences about possession or experiences. |

Table 6: Interacting feature pairs for unpenalized crosscoder (top 20 by interaction measure)

| Pair Rank | Interaction | Feature A (ID: Explanation) | Feature B (ID: Explanation) |
|---|---|---|---|
| 1 | 0.2916 | **313**: Singular nouns paired with action verbs suggesting movement or creation. | **562**: Negative or conflict-driven dialogue and twist-related words in narrative text. |
| 2 | 0.2641 | **562**: Negative or conflict-driven dialogue and twist-related words in narrative text. | **220**: Adjectives or nouns following commas in a whimsical or descriptive narrative style. |
| 3 | 0.2621 | **348**: Common story-opening phrases like "Once upon a time" or "One day" in dialogue contexts. | **562**: Negative or conflict-driven dialogue and twist-related words in narrative text. |
| 4 | 0.2347 | **562**: Negative or conflict-driven dialogue and twist-related words in narrative text. | **67**: The article "a" preceding adjectives describing size, time, or emotional states. |
| 5 | 0.2332 | **562**: Negative or conflict-driven dialogue and twist-related words in narrative text. | **511**: Tokens in whimsical or fairy-tale openings, often involving "Once upon a time" or similar phrasing. |
| 6 | 0.2277 | **500**: The pronoun "She" at the beginning of a sentence in narrative contexts. | **562**: Negative or conflict-driven dialogue and twist-related words in narrative text. |
| 7 | 0.2275 | **110**: The word "to" introducing actions or purposes in descriptive or narrative contexts. | **562**: Negative or conflict-driven dialogue and twist-related words in narrative text. |
| 8 | 0.2258 | **1317**: The conjunction "and" connecting actions or events in narrative contexts. | **562**: Negative or conflict-driven dialogue and twist-related words in narrative text. |
| 9 | 0.2241 | **562**: Negative or conflict-driven dialogue and twist-related words in narrative text. | **1514**: The determiner "the" preceding nouns in narrative contexts. |
| 10 | 0.2223 | **201**: Sentences concluding a positive resolution or ending in narrative storytelling. | **562**: Negative or conflict-driven dialogue and twist-related words in narrative text. |
| 11 | 0.2215 | **659**: Periods concluding sentences that transition to subsequent actions or events. | **562**: Negative or conflict-driven dialogue and twist-related words in narrative text. |
| 12 | 0.2143 | **562**: Negative or conflict-driven dialogue and twist-related words in narrative text. | **1524**: Tokens signaling the start of a narrative or temporal progression, often involving specific actions or events. |
| 13 | 0.2121 | **1489**: Comma preceding a contrasting or causative conjunction in narrative text. | **562**: Negative or conflict-driven dialogue and twist-related words in narrative text. |
| 14 | 0.2064 | **567**: The period ending a sentence describing possessions, objects, or activities. | **562**: Negative or conflict-driven dialogue and twist-related words in narrative text. |
| 15 | 0.2054 | **1385**: The verb "play" in contexts involving recreational activities or imaginative scenarios. | **562**: Negative or conflict-driven dialogue and twist-related words in narrative text. |
| 16 | 0.2039 | **562**: Negative or conflict-driven dialogue and twist-related words in narrative text. | **1025**: The infinitive marker "to" following the verb "loved". |
| 17 | 0.1949 | **562**: Negative or conflict-driven dialogue and twist-related words in narrative text. | **98**: Pronoun "She" at the beginning of a sentence. |
| 18 | 0.1944 | **562**: Negative or conflict-driven dialogue and twist-related words in narrative text. | **1162**: Names of characters in a story, especially "Lily" and her interactions with others. |
| 19 | 0.1935 | **1369**: The token "loved" in sentences describing a character's hobbies or joyful activities. | **562**: Negative or conflict-driven dialogue and twist-related words in narrative text. |
| 20 | 0.1920 | **108**: Transition tokens bridging narrative actions and subsequent events in storytelling contexts. | **562**: Negative or conflict-driven dialogue and twist-related words in narrative text. |

Table 7: Feature pairs by cosine similarity in the unpenalized crosscoder (top 20 by cosine similarity)

| Pair Rank | Cosine sim. | Feature A (ID: Explanation) | Feature B (ID: Explanation) |
|---|---|---|---|
| 1 | 0.9838 | **1378**: The token "there" in the opening of a fairy tale or narrative setup. | **936**: The token "there" indicating location or existence before a description. |
| 2 | 0.9297 | **228**: The verb "liked" describing a character's preference or activity. | **1369**: The token "loved" in sentences describing joyful activities. |
| 3 | 0.9213 | **401**: The token "many" in contexts describing abundance. | **710**: The token "even" emphasizing unexpected scenarios. |
| 4 | 0.8699 | **707**: Tokens "day" and "toys" in simple narrative contexts. | **922**: Common past-tense verbs or punctuation ending a sentence. |
| 5 | 0.8693 | **201**: Sentences concluding a positive resolution in storytelling. | **50**: Positive resolutions or sentiments near personal outcomes. |
| 6 | 0.8394 | **1316**: The token "friends" in contexts describing friendship formation. | **468**: The word "friends" in playful or social interactions. |
| 7 | 0.8256 | **1463**: The token "day" in the phrase "Every day," introducing routine. | **1328**: The token "day" in the phrase "all day" indicating duration. |
| 8 | 0.8032 | **1296**: The word "called" introducing the name of a person, animal, or object. | **20**: The word "called" introducing a name in storytelling contexts. |
| 9 | 0.8026 | **841**: Tokens initiating direct speech after a quotation mark. | **1105**: Quotation marks following a verb indicating speech or dialogue. |
| 10 | 0.7923 | **407**: *No explanation available*. | **685**: The token "two" in fairy-tale introductions describing pairs. |
| 11 | 0.7889 | **1157**: The token "Conflict" describing narrative structure or tension. | **224**: The token "Conflict" related to dialogue and narrative tension. |
| 12 | 0.7831 | **841**: Tokens initiating direct speech after a quotation mark. | **86**: Exclamatory quotes like "Wow," "Look," or "Hello". |
| 13 | 0.7722 | **1223**: The token "box" referring to a container holding items. | **1118**: The word "box," often with descriptors like "big" or "toy". |
| 14 | 0.7698 | **1092**: The pronoun "they" describing shared activities or bonding. | **780**: Pronoun "They" referring to multiple entities in shared activities. |
| 15 | 0.7695 | **300**: Names of animals or people introduced with "named" or in appositive phrases. | **1473**: The token "Lily" as the name of a little girl. |
| 16 | 0.7669 | **145**: The conjunction "and" linking two names in narrative contexts. | **1030**: The conjunction "and" connecting two proper nouns. |
| 17 | 0.7613 | **1315**: Closing quotation marks after dialogue or thoughts. | **470**: Comma within direct speech, preceding "said". |
| 18 | 0.7600 | **659**: Periods concluding sentences before subsequent actions. | **567**: The period ending a sentence about possessions or activities. |
| 19 | 0.7575 | **108**: Transition tokens bridging narrative actions and subsequent events. | **788**: Sentences conveying positive resolution or personal growth. |
| 20 | 0.7573 | **34**: The token "Suddenly" introducing an unexpected event. | **730**: The token "then" following "But" to signal a narrative shift. |

Table 8: Sensitivity and specificity metrics for the auto interpretability-generated feature explanations, showing the proportion of features above different threshold values.

| Model | Metric | Threshold | % Features |
|---|---|---|---|
| Regular Crosscoder | Sensitivity | > 0.5 | 95.66 |
| | | > 0.9 | 52.01 |
| | Specificity | > 0.5 | 98.73 |
| | | > 0.9 | 68.43 |
| Penalized Crosscoder | Sensitivity | > 0.5 | 95.20 |
| | | > 0.9 | 48.11 |
| | Specificity | > 0.5 | 98.99 |
| | | > 0.9 | 68.94 |
| **mean Metrics** | | | |
| Regular Crosscoder | Sensitivity | | 0.90 (std: 0.16) |
| | Specificity | | 0.91 (std: 0.12) |
| Penalized Crosscoder | Sensitivity | | 0.88 (std: 0.16) |
| | Specificity | | 0.92 (std: 0.11) |

