# OpenReview forum: "Interactions between crosscoder features: a compact proofs perspective"
_ICLR.cc/2026/Conference — Submitted to ICLR 2026_

### Official Review · Reviewer_3w9m · 2025-10-31

**Soundness:** 3
**Presentation:** 2
**Contribution:** 2
**Rating:** 4
**Confidence:** 4

**Summary:**

This paper proposes a method to recursively decompose crosscoder reconstruction error into inter-layer propagation terms and "feature transition error" and derives a computable interaction metric on MLP layers to quantify the error caused by non-dominant features when interpreted by a single dominant feature. The authors also demonstrate that using this metric as a regularization term to train the crosscoder significantly improves the "dominant feature share" and conduct a series of experimental validations.

**Strengths:**

1、Provides a hierarchical recursive error decomposition framework and proposes a practically computable metric, bridging formal analysis and applications.

2、The interaction metric on MLPs has explicit computational steps and can be directly used as a training regularization term or analysis tool.

3、The work evaluates multiple uses of the metric，showing diverse potential applications.

**Weaknesses:**

1、The steps of applying a coarse upper bound to ReLU and scaling using norms may result in a loose upper bound for the final error. The lack of formalization of attention and LayerNorm limits the completeness of the theory.

2、The experimental section only presents the results without further analysis and interpretation.

3、Several symbols and assumptions ($g^l_v(u_v), \hat{e}_v, e_v, W^{l}\_{in}$) are not clearly defined, making the logical chain of equations (6)–(9) confusing to read. Clearer symbols and explanations are needed.

**Questions:**

1、The core decomposition is based on a single dominant feature. Are there scenarios where multiple features dominate? How should it be extended in these scenarios?

2、The method validation is mainly conducted on small/medium-sized models in the TinyStories style. How does it perform in other task scenarios?

---

> ### Author Response · Authors · 2025-11-24
>
> We thank the reviewer for taking the time to review our paper and for recognizing our formal analysis and empirical applications.
>
> In the revised manuscript, we have added experiments that show robustness of our results in the Pythia family of models, across three orders of magnitude of model sizes. We have also corrected typos and improved the clarity of the manuscript. We believe the resulting manuscript is significantly stronger, and are grateful to the reviewer for their input.
>
> We provide specific responses below:
>
> 1. **The steps of applying a coarse upper bound to ReLU and scaling using norms may result in a loose upper bound for the final error. The lack of formalization of attention and LayerNorm limits the completeness of the theory.**
>
> We agree that providing explicit formulas for attention and Layernorm are important areas for further work, and we provide initial results for attention in Appendix C.
>
> 2. **The experimental section only presents the results without further analysis and interpretation.**
>
> We provide a discussion in section 4.2 detailing why it is desirable to obtain computationally sparse crosscoders, given in the response to point 3 of reviewer yDNG. In section 4.3 we interpret the clustering result to mean that "Second, we show that we can use the interaction metric to find larger scale structure by clustering features—ultimately this could show us which combinations of features should combine into feature circuits. "
>
> We are happy to add additional discussion with the additional space provided in the camera-ready version.
>
> 3. **Several symbols and assumptions  are not clearly defined, making the logical chain of equations (6)–(9) confusing to read. Clearer symbols and explanations are needed.**
>
> We have added this clarification to the main text:
>
> >Let $W^{l}\_{in}, b^l\_{in} W^{l}\_{out},b^l\_{out}$ denote the weight matrices and bias vectors mapping into and out of the MLP activation function at layer $l$. As in \cref{sec:crosscoder_overview}, let $W^l\_{enc}, b^l\_{enc}; W^l\_{dec}, b^l\_{dec}$ denote the weight matrices and bias vector for the encoding and decoding respectively. Let $\hat{e}\_v$ denote the unit vector corresponding to feature $v$.
>
> 4. **The core decomposition is based on a single dominant feature. Are there scenarios where multiple features dominate? How should it be extended in these scenarios?**
>
> We have added appendix section B.3 and B.4 clarifying which extensions are possible, and proposing a generalization based on Shapley-Taylor Interaction Indices. The choice made in the paper is the simplest possible - at the cost of potentially unacceptable losses to reconstruction error. Since we empirically verify a favourable trade-off, this choice is sufficient in our work. Further decompositions are an important area for further work.
>
> 5. **The method validation is mainly conducted on small/medium-sized models in the TinyStories style. How does it perform in other task scenarios?**
>
> We have added clearer referencing to Fig 9 Appendix E.3 where we provided data for crosscoders up to an expansion factor of 8x. We also provide additional experiments on Pythia models across three orders of magnitude, and crosscoders trained on them up to expansion factors of 4x in Fig 8 Appendix E.3. We see strikingly similar trade-offs across model and crosscoder sizes.

---

### Official Review · Reviewer_ZdYD · 2025-10-31

**Soundness:** 3
**Presentation:** 2
**Contribution:** 2
**Rating:** 4
**Confidence:** 2

**Summary:**

By bounding the layer-wise reconstruction error and decomposing the transition term into a sum of single-feature contributions plus an interaction residual, the paper provides: (1) an in-principle procedure to derive compact proofs, and (2) a closed-form metric for quantifying feature interactions in MLPs. Empirically, the paper shows that training with an interaction penalty raises the dominant-feature share from around 30\% to 60\% without increasing reconstruction error.

**Strengths:**

1. Using the metric as a penalty increases the dominant-feature share dramatically with modest reconstruction cost (Fig. 1a) and materially improves dominant-only ablation fidelity (Fig. 2a). This provides evidence that the interaction metric captures meaningful interactions.
1. The closed-form interaction metric is simple to compute and linear in the number of active features per token and neuron, which contrasts the exponential difficulty of calculating STII.
1. The paper examines interactions, performs layer-wise ablations, explores clustering of interacting features, and probes a sleeper-agent setting where interactions plausibly matter. The empirical narrative is cohesive.

**Weaknesses:**

1. As the authors admit, the in-principle procedure given in Section 3 depends on loose triangle inequality, Lipschitz bounds, and the accumulation of errors across layers. The explicit closed-form result is presently given for the MLP layer only, which limits practical generalizability; the overall procedure is broader in scope but incomplete for attention and LayerNorm.
1. Application III's anomaly detection results rely on an interaction metric derived only for MLP layers (Eq. 9), while attention and layer norm are not formally covered. Appendix C sketches a sparse Q/K feature-interaction decomposition, but OV and LN remain unresolved and a complete compact proof is not yet available. Consequently, sleeper behaviors mediated by attention pathways may be under-detected by this MLP-based signal.
1. The method assumes a single "dominant" contributor per neuron, which can be brittle under near-ties. Would you clarify whether the top-1 choice is essential or whether it can be generalized to a few-hot variant?
1. Clarity has room for improvement:
    1. The term "proof" appears to mean a verification procedure rather than a formal theorem proof. For clarity to readers outside the compact-proofs community, please define this usage explicitly early on (excluding the fixed phrase "Compact proof"). In line 119, the authors mention "the full details of the proof," which seems to refer to the full details of the reduction and bounds derivation, rather than a formal theorem proof.
    1. The derivation of the interaction metric in Section 3 contains several notational inconsistencies and undefined symbols. Please clarify: $g_v^l(u_v)$, $g^l(u_v)$, $g_v^l(u)$ all appear but seem to refer to the same object; $\varepsilon$, $\hat e_v$, $W_{out}^l$, $b_{out}^l$ are used but not defined; what is the index $i$ appearing in Eq. (9)?
    1. There are minor typos throughout the paper. Some examples are as follows:
         - line 17: "inteaction" $\rightarrow$ "interaction"
         - line 112: "fronteir" $\rightarrow$ "frontier"
         - line 195: "the the"
         - line 206: "interaction metric Eq. (6)" $\rightarrow$ "interaction metric in Eq. (9)"
         - line 319: "implements parallelizes"
         - line 351: "investige" $\rightarrow$ "investigate"
         - line 403: "Anomlay" $\rightarrow$ "Anomaly"
         - line 414: "occurences" $\rightarrow$ "occurrences"
         - line 486: "insured" $\rightarrow$ "ensured"
         - line 496: "publically" $\rightarrow$ "publicly"
         - line 499: "where" $\rightarrow$ "were"

**Questions:**

1. What concrete obstacles prevent a closed-form interaction metric for attention and LayerNorm?
1. How sensitive are your results to the top-1 dominant assumption?

---

> ### Author Response · Authors · 2025-11-24
>
> We thank the reviewer for taking the time to review our paper and for acknowledging our experimental narrative and the advantages of our scheme over the STII.
>
> We provide responses to specific points below:
>
> 1. **As the authors admit, the in-principle procedure given in Section 3 depends on loose triangle inequality, Lipschitz bounds, and the accumulation of errors across layers. The explicit closed-form result is presently given for the MLP layer only, which limits practical generalizability; the overall procedure is broader in scope but incomplete for attention and LayerNorm.**
>
> We agree with the important point that we do not yet have explicit expressions for every layer in the network - this an important direction for further work.
>
> 2. **Application III's anomaly detection results rely on an interaction metric derived only for MLP layers (Eq. 9), while attention and layer norm are not formally covered. Appendix C sketches a sparse Q/K feature-interaction decomposition, but OV and LN remain unresolved and a complete compact proof is not yet available. Consequently, sleeper behaviors mediated by attention pathways may be under-detected by this MLP-based signal.**
>
> We agree that we are unable to detect attention-mediated sleepers, and that extending our proof to attention would open this important application.
>
> 3. **The method assumes a single "dominant" contributor per neuron, which can be brittle under near-ties. Would you clarify whether the top-1 choice is essential or whether it can be generalized to a few-hot variant?**
>
> We have added appendix section B.3 and B.4 clarifying which extensions are possible, and proposing a generalization based on Shapley-Taylor Interaction Indices. The choice made in the paper is the simplest possible - at the cost of potentially unacceptable losses to reconstruction error. Since we empirically verify a favourable trade-off, this choice is sufficient in our work. Further decompositions are an important area for further work.
>
> 4. **The term "proof" appears to mean a verification procedure rather than a formal theorem proof. For clarity to readers outside the compact-proofs community, please define this usage explicitly early on**
>
> We provide a formal definition for "Compact proof" in Appendix B:
>
> >Formally, we define a \textit{compact proof} following \cite{gross2024compactproofs}.
> >Let the model $ \mathcal{M} : X \to Y $ be a map from the set of inputs $X$ to outputs $Y$ and let $L$ be the set of labels associated to each input. For $\mathcal{D}$ a probability distribution over (label,input) pairs, and $f:L \times Y \to \mathbb{R}$ a scoring function (typically the accuracy or loss) define $b$ as a bound of the expectation value of the scoring function over $\mathcal{D}$:
> >
> >$$ b\ \geq \mathbb{E}_{\mathcal{D}} [f(l,\mathcal{M}(x))] $$
> >
> >A compact proof is then a proof $Q$ establishing a bound $b$ and a computational verifier $C$ whose runtime measures the compactness of the proof. In this paper, the proof $Q$ is the bound established by the crosscoder on the model
> %  (Eq (4) of section 7 of the Supplement)
> and the verifier $C$ is the computational trace which evaluates the error terms of the bound. In our case, the bound $b$ is the bound on the output error - that is the difference between the final layer residual stream activations $a^N(x)$ and the decoding to the final layer $W^N(u(x))$.
>
> 5. **The derivation of the interaction metric in Section 3 contains several notational inconsistencies and undefined symbols. Please clarify: $ g^l\_v(u_v), g^l(u\_v), g^l_v(u) $all appear but seem to refer to the same object; $\varepsilon, \hat{e}\_v, W^l\_{out}, b^l\_{out} are used but not defined; what is the index appearing in Eq. (9)?**
>
> We standardize the notation for $g$, define the relevant symbols and clarify that $i$ is the (dominant) feature index in the main text.
>
> 6. **There are minor typos throughout the paper.**
>
> Fixed - thank you.
>
> 7. **What concrete obstacles prevent a closed-form interaction metric for attention and LayerNorm?**
>
> The MLP-preactivation is linear in the features, whereas the Attention preactivation is quadratic. This makes is difficult to identify the "default" behaviour. In Appendix C, we describe that the difficulty of dealing with attention is that "we need to take into account positional variation, combine values across multiple sequence positions, and deal with queries, keys and values mixing together contributions from many different features. " LayerNorm has a complicated functional form, and also depends on sequence position, and so it is similarly difficult to identify a non-interacting "default"
>
> 8. **How sensitive are your results to the top-1 dominant assumption?**
>
> We give sensitivity data in Fig 2 and Fig 7. The effect of ablating the second dominant feature falls as we increase the penalty. At $\lambda=1000$, the effect is close to ablating a random feature. We refer to the figures for further data.

---

### Official Review · Reviewer_yDNG · 2025-10-31

**Soundness:** 3
**Presentation:** 3
**Contribution:** 3
**Rating:** 6
**Confidence:** 2

**Summary:**

This paper builds on previous work showing that crosscoders can be used as a way to generate a compact "proof" of model capabilities, showing an error bound on LLM capability using a crosscoder. While this error bound is too large to be useful in practice, this is used to derive a loss term that can minimize interactions between features in crosscoder. The authors then show that this penalized cross-coder finds interpretable features using auto-interpretability, and demonstrate that the remaining interactions between features be useful to debug strange behavior in models, such as finding sleeper agent behavior from previous work.

**Strengths:**

- The paper offers a lot of theoretical grounding. I do not have a mathematics or theory background, so cannot judge how accurate the theory is, but it seems reasonable to me.
- The paper trains crosscoders on a real LLM showing that the technique can scale past toy models, and shows interesting interpretability benefits like uncovering sleeper agents from previous work.
- Characterizing interactions between features feels like a good intermediate step between just detecting features and locating full circuits.
- The paper provides supporting evidence on LLM crosscoders via auto-interp results and case-studies.
- It sounds like this opens up a lot of avenues for future work that builds on the findings in this paper.

**Weaknesses:**

This paper is a bit out of my domain as I do not have a theory background, so I struggle to follow a lot of the ideas in the paper. Where I do not understand the theory I give the authors the benefit of the doubt. That being said:

- The crosscoders trained in this work are not very big, having an expansion factor of 2 would be very tiny for an SAE, for example. Crosscoders are very expensive to train, though, so maybe there is no way around this.
- The bounds from the proof are too large to be useful, so I'm not sure what practical benefit the framing of crosscoders as providing proofs has.
- The loss proposed in the paper seems to encourage crosscoder features to map to a single LLM neuron (if I understand things correctly). I don't understand why this is an inherently beneficial property.

**Questions:**

- L100: "We set the decoder bias to zero to avoid assigning it to features" what does it mean to assign the decoder bias to features?
- In section 2, where are the inputs and outputs to the crosscoder coming from? I assume the input is the residual stream at layer $l$, but what is the output? Is the the MLP output or the residual steam of the next layer?
- The paper repeatedly references sections of the "SM" for "Supplementary Material". I think this is referring to the Appendix, as I do not see any supplementary material provided in the submission, but I cannot find the sections referenced in the paper in the appendix. It would help if the authors directly link to the section in the appendix that is referenced in the main body rather than just referring to  the "SM".
- It sounds like the extra loss penalty tries to force crosscoder features to align exactly with one neuron, if I'm understanding this correctly. Why is this desirable?
- The paper picks a $g(x)$ that tries to make each feature align with a single neuron, but if I understand correctly, $g(x)$ can be picked arbitrarily. What other choices of $g(x)$ are worthwhile to use? Or is the $g(x)$ used in the paper the only choice that makes sense?

### Minor issues / formatting
- L17: "inteaction" should be "interaction"
- L66: typo in "the Section 7 of theSupplementary Material"
- L73: typo "meaninfgul"
- L75: "sleeper agents Hubinger et al." seems like it's missing a word

---

> ### Author Response · Authors · 2025-11-24
>
> We thank the reviewer for carefully reading our paper and for their insightful suggestions.
>
> In our revision, we clarified that although the proof procedure we provide cannot yet be extended to the full model, it provides a roadmap for further work to do so. We provide additional experiments for expanded crosscoders sizes for both TinyStories and Pythia models spanning three orders of magnitude. We also clarify that our proof allows extension beyond single features, and highlight that this is an important direction for further work. We believe these changes have strengthened the manuscript and are grateful to the reviewer for pointing them out.
>
> We provide specific responses below:
>
> 1. **The crosscoders trained in this work are not very big, having an expansion factor of 2 would be very tiny for an SAE, for example. Crosscoders are very expensive to train, though, so maybe there is no way around this.**
>
> We have added clearer referencing to Fig 9 Appendix E.3 where we provided data for crosscoders up to an expansion factor of 8x. We also provide experiments on Pythia models across three orders of magnitude, and crosscoders trained on them up to expansion factors of 4x in Fig 8 Appendix E.3. We see strikingly similar trade-offs across model and crosscoder sizes.
>
> 2. **The bounds from the proof are too large to be useful, so I'm not sure what practical benefit the framing of crosscoders as providing proofs has.**
>
> We have clarified in the main text that:
>
> >“We emphasize that although we cannot obtain non-vacuous bounds for the full model, the bounds are not vacuous in a given MLP layer - as shown in Fig. 2d. We hence consider the interaction metric and its applications to be the main contribution of this work.
> > The proof procedure we show here, however, is general and can be extended to other layers. It thus provides a roadmap towards
> > formally verfiying how much of a model’s behaviour a given crosscoder can explain.”
>
> 3. **The loss proposed in the paper seems to encourage crosscoder features to map to a single LLM neuron (if I understand things correctly). I don't understand why this is an inherently beneficial property.**
>
> >This is desirable for two reasons. First, it allows us to obtain a better approximation for the crosscoder on the basis of a single feature (per datapoint and per neuron). This means we can more verify a bound on the crosscoder reconstruction by only computing the dominant feature. Second, this reduces the error from the non-linearity between layers (RHS of Eq 6) since we do not need to consider the, in general exponentially many, interactions. Given that we approximate the MLP as the dominant feature + its interactions with other features, this effectively means we reduce the interaction. This is in turn beneficial for mechanistic anomaly detection, since we only need to monitor single feature that compose linearly.
>
> 4. **We set the decoder bias to zero to avoid assigning it to features" what does it mean to assign the decoder bias to features?**
>
> This follows from the fact that the decoder bias is constant for every feature present at a given latent in the residual stream.
> Hence the reconstruction of the residual stream neuron is the weighted sum of features + a bias term which is constant for all features at a given latent. This means that if we want to assign the preactivation to the features, we would need to assign the bias to the features in some way, with the natural choice being to add a term $b^l\_{in}/h$ to every feature. Explicitly, the preactivation of a neuron $k$ at layer $l$ can be written in terms of the features as:
>
> $$ z^l\_k=W^l\_{in}(W^l\_{dec}u_v\hat{e}\_v+b^l\_{dec})+b^l\_{in} $$
>
> If we want to consider $ z^l\_k $ as a function of the $u_v$ with full reconstruction when all features are present, we must assign both b^l\_{dec}) to the features which introduces an ambiguity. We can avoid this by setting it to zero.
>
> 5. **The paper repeatedly references sections of the "SM" for "Supplementary Material..."**
>
> We have updated all references to explicit section references - thank you.
>
> 6. **It sounds like the extra loss penalty tries to force crosscoder features to align exactly with one neuron, if I'm understanding this correctly. Why is this desirable?**
>
> Addressed in point 3.
>
> 7. **The paper picks a $g(x)$ that tries to make each feature align with a single neuron, but if I understand correctly, can be picked arbitrarily. What other choices of are worthwhile to use? Or is the used in the paper the only choice that makes sense?**
>
> We have added appendix section B.3 and B.4 clarifying which extensions are possible, and proposing a generalization based on Shapley-Taylor Interaction Indices. The choice made in the paper is the simplest possible - at the cost of potentially unacceptable losses to reconstruction error. Since we empirically verify a favourable trade-off, this choice is sufficient in our work. Further decompositions are an important area for further work.

---

### Official Review · Reviewer_edi4 · 2025-11-01

**Soundness:** 3
**Presentation:** 2
**Contribution:** 3
**Rating:** 4
**Confidence:** 3

**Summary:**

This work proposes the interaction metric and showcases how it can be applied to train computationally sparse crosscoders, capture semantically meaningful interactions and be leveraged to for mechanistic anomaly detection.

**Strengths:**

1. The paper proposes a novel measure and showcases how this metric can be used to train computationally sparse crosscoders and capture semantically meaningful interactions.

2. The work provides supporting evidence for its claim.

**Weaknesses:**

1. The structure of the paper and the density of the sections make it difficult to follow the contributions, both the abstract and introduction poise the compact proof as the major contribution. Then the conclusion of section 3 and the limitation section highlight that the proposed proof is unlikely to lead to non-vacuous bounds and impractical for larger models. Sections 4, 5 and 6 then focus on the Interaction Metric which is presented as a byproduct of the compact proof in section 3.

2. Section 6 is short and appears rushed, especially when compared with section 3 and 4.

3. Terms in section 3 that are not introduced, $W^{l}{in}$ $W^{l}{out}$, $\hat{e}_{v}$ when used for the first time, breaking the reading flow.

4. General typos: 319 “implements parallelizes”, 320 “to to”.

5. The analysis appears to have been conducted on a small number of models, whether the proposed metric would work under different settings remains to be shown.

**Questions:**

1. I could not find the fidelity metric F in Bricken 2023, which I believe is the following paper https://transformer-circuits.pub/2023/monosemantic-features/index.html, if CMD+f for Fidelity I can’t find it. Link obtained from https://www.anthropic.com/research/towards-monosemanticity-decomposing-language-models-with-dictionary-learning, for the paper Bricken 2023 2nd reference in the reviewed paper., could not find “reinsert”, “insert” either, but I could find “We often normalize this by dividing by the difference in loss between the baseline transformer’s performance and its performance after ablating the MLP layer. This gives us a fraction of the MLP’s loss contribution that is explained by our transformer. However, the performance with an abated MLP may be an especially bad baseline, so this percentage is considered an overestimate.” (Bricken 2023)]]

2. The proposal gets diluted, is the proof the main proposal? Is the interaction metric the main proposal?

3. Equation (12) what is T?

4. Is there a link between the F in equations (11) an (12), it might be clearer to use a different letter if they are unrelated?

5. Could the authors provide an example of the confusion matrix mentioned in section 4.3 loc [432,433]
[[Equation is (11) could be clarified, what is the difference between $L_0$ and $_{ablated}$?]]

---

> ### Author Response · Authors · 2025-11-24
>
> We thank the reviewer for their careful reading of our paper, and for recognizing the novelty of our measure and the strength of the evidence.
>
> We appreciate the feedback on improving the clarity of the text, and have implemented the suggested improvements on the clarity of the paper (detailed below), and conducted additional experiments to demonstrate that our findings are robust across three orders of magnitude of model sizes and on The Pile dataset (Fig 5, Appendix B in revised manuscript). We believe this has significantly strengthened the clarity of our presentation and the generalizability of our results.
>
> 1. **The structure of the paper and the density of the sections make it difficult to follow the contributions,...**
>
> We implement these helpful suggestion by adding a paragraph in section 2 clarifying the relative importance of our contributions:
>
> >“We emphasize that although we cannot obtain non-vacuous bounds for the full model, the bounds are
> >not vacuous in a given MLP layer - as shown in Fig. 2d. We hence consider the interaction metric and
> >its applications to be the main contribution of this work. The proof procedure we show here, however,
> >is general and can be extended to other layers. It thus provides a roadmap towards formally verfiying
> >how much of a model’s behaviour a given crosscoder can explain.”
>
> 2. **Section 6 is short and appears rushed, especially when compared with section 3 and 4.**
>
> We unfortunately had to shorten this section due to the length limit, but will expand with additional details using the space available in the camera-ready version.
>
> 3. **Terms in section 3 that are not introduced, $ W^l\_{out}, W^l\_{in}, \hat{e}\_{v}  $ when used for the first time, breaking the reading flow.**
>
> We have clarified:
>
> > Let $ W^{l}\_{in}, b^l\_{in} W^{l}\_{out},b^l\_{out}$ denote the weight matrices and bias vectors mapping into and out of the MLP
> > activation function at layer $l$... Let Let $\hat{e}\_v$ denote the unit vector corresponding to feature $v$.
>
> 4. **General typos: 319 “implements parallelizes”, 320 “to to”.**
>
> Fixed, thank you.
>
> 5. **The analysis appears to have been conducted on a small number of models, whether the proposed metric would work under different settings remains to be shown.**
>
> We have provided in Fig 5, Appendix B of the revised manuscript results across three orders of magnitude for Pythia models - from 13M to 1B -  trained on the Pile where we document strikingly similar trade-offs to those found in our mainline TinyStories model.
>
> 6. **I could not find the fidelity metric F in Bricken 2023**
>
> We have added a more explicit reference to the fidelity metric we use:
>
> > define the fidelity $ \Phi $ as the \textit{loss recovered} (Eq (5) of [1]) relative to a baseline of zero ablation in the MLP of the same
> > layer
>
> [1] - *Improving dictionary learning with gated sparse autoencoders, 2024, Rajamanoharan et al, arXiv 2404.16014*
>
> 7. **The proposal gets diluted, is the proof the main proposal? Is the interaction metric the main proposal?**
>
> We address this in the response to point 1 above.
>
> 8. **Equation (12) what is T?**
>
> We thank the reviewer for this clarification. We use the notation of the reference work. We have clarified that:
>
> > We consider a target function $F$ with argument given by the full set of active features (by convention denoted as) $T=\
> >{u_1,...,u_h\}$ at each neuron $k$...
>
> 9. **Is there a link between the F in equations (11) an (12), it might be clearer to use a different letter if they are unrelated?**
>
> We change the notation for the fidelity $ F $ to $ \Phi $ in order to remain consistent with the notation used for the reference work in Eq 12.
>
> 10. **Could the authors provide an example of the confusion matrix mentioned in section 4.3 loc [432,433]**
>
> We provide the explicit confusion matrix in Fig 11.
>
>  11. **Equation (11) could be clarified, what is the difference between $\mathcal{L}\_0 and \mathcal{L}\_{ablate} ?**
>
> We clarify in the main text that:
>
> >where $ \mathcal{L}\_{ablate} $ is the result of ablating the target features, $ \mathcal{L\_M} $ is the original model loss, and
> > $\mathcal{L}\_0$ is the result of zero ablating \textit{all} features in the target layer.

---

### Meta-Review · Area_Chair_rrfH · 2026-01-07

**Summary:**

This work focuses on the crosscoder, demonstrating that decomposing reconstruction error into feature-specific components and an interaction residual can be used to derive both a compact proof of model performance and a formal measure of feature interactions. While this error bound is too large to be useful in practice, this is used to derive a loss term that can minimize interactions between features in crosscoder. The authors then show that this penalized cross-coder finds interpretable features using auto-interpretability, and demonstrate that the remaining interactions between features be useful to debug strange behavior in models, such as finding sleeper agent behavior from previous work.

**Reviewer Concerns:**

Several shared concerns emerged among the reviewers: (1) the derivation relies on extremely loose upper bounds (e.g., via the triangle inequality, Lipschitz bounds, and error accumulation across layers), resulting in estimates that are too large to serve as practical formal proofs; (2) the current derivation is restricted to MLP layers, and the lack of formalization for Attention and LayerNorm significantly limits the framework’s generalizability to modern Transformer architectures; (3) there is a persistent mismatch between the paper’s framing (which emphasizes formal proofs) and the applications in later sections; and (4) Section 6 lacks sufficient detail.

The authors addressed some of these issues by adding preliminary discussions regarding attention in the appendix and clarifying the relative importance of their various contributions. Nevertheless, significant concerns remain, particularly regarding the looseness of the upper bounds and the lack of clarity on how the theoretical derivations in Section 3 directly relate to the subsequent practical applications.

**Reviewer Scores:**

While some concerns have been addressed, given the skepticism regarding 'loose upper bounds,' scaling issues, and the tenuous connection between the theoretical bounds and the subsequent applications, several reviewers are likely to maintain their rating of 4.

---

### Decision · Program_Chairs · 2026-01-26

Reject